# Enhancing Semi-supervised Learning with Zero-shot Pseudolabels

**Jichan Chung**                                                    *jichan3751@berkeley.edu*
*Department of EECS*
*University of California, Berkeley*

**Irene Y. Chen**                                                    *iychen@berkeley.edu*
*Department of EECS*
*University of California, Berkeley*

**Reviewed on OpenReview:** *https://openreview.net/forum?id=WB05Doi29V*

## Abstract

The high cost of data labeling presents a major barrier to deploying machine learning systems at scale. Semi-supervised learning (SSL) mitigates this challenge by utilizing unlabeled data alongside limited labeled examples, while the emergence of foundation models (FMs) offers powerful zero-shot capabilities that can further reduce labeling cost. However, directly fine-tuning large FMs is often impractical in resource-constrained settings, and naïvely using their pseudo-labels for unlabeled data can degrade performance due to its unreliablity or domain mismatch with target task. In this work, we introduce ZeroMatch, a novel SSL framework that integrates knowledge distillation with consistency-based learning to jointly leverage labeled data, unlabeled data, and pseudo-labels from FMs. ZeroMatch trains a compact student model and access FMs only through inference services, making it suitable for low-resource environments such as personal devices with limited compute. Experiments on six vision and language classification benchmarks show that ZeroMatch consistently outperforms standard SSL and zero-shot augmented methods, demonstrating its effectiveness and robustness across a range of foundation model qualities.[1]

## 1 Introduction

The growing scale of machine learning applications has made data labeling costs a critical bottleneck in deploying ML systems (Northcutt et al., 2021; Sun et al., 2017; Shen et al., 2024). Semi-supervised learning (SSL) addresses this challenge by leveraging unlabeled data alongside limited labeled examples (Tarvainen & Valpola, 2017). Traditional SSL approaches like pseudo-labeling and consistency regularization have demonstrated strong performance across domains, particularly in computer vision and natural language processing (Sohn et al., 2020; Laine & Aila, 2022; Tarvainen & Valpola, 2017).

In parallel, the emergence of foundation models (FMs) has opened new opportunities for reducing reliance on labeled data. These large-scale pre-trained models exhibit strong zero-shot capabilities, enabling them to generalize to novel tasks without requiring task-specific fine-tuning (Brown et al., 2020; Liang et al., 2022). To this end, recent efforts have explored integrating foundation models into the SSL pipeline. Proposed strategies include fine-tuning foundation models with labeled and unlabeled data (Shi et al., 2023; Zhang et al., 2024; Gan & Wei, 2024), using zero-shot predictions as pseudo-labels (Hegselmann et al., 2023; Nam et al., 2023), and distilling knowledge from foundation models into smaller student models (Yang et al., 2025; Shi et al.; Vemulapalli et al., 2023; Zhao et al., 2023; Jiang et al., 2023).

We motivate our problem setting with a practical on-device training scenario: a user with a personal modest-sized dataset and limited computational resources (e.g., a single GPU), with access to foundation model

---

[1]Experiment codes are available in `https://github.com/jichan3751/zeromatch`.

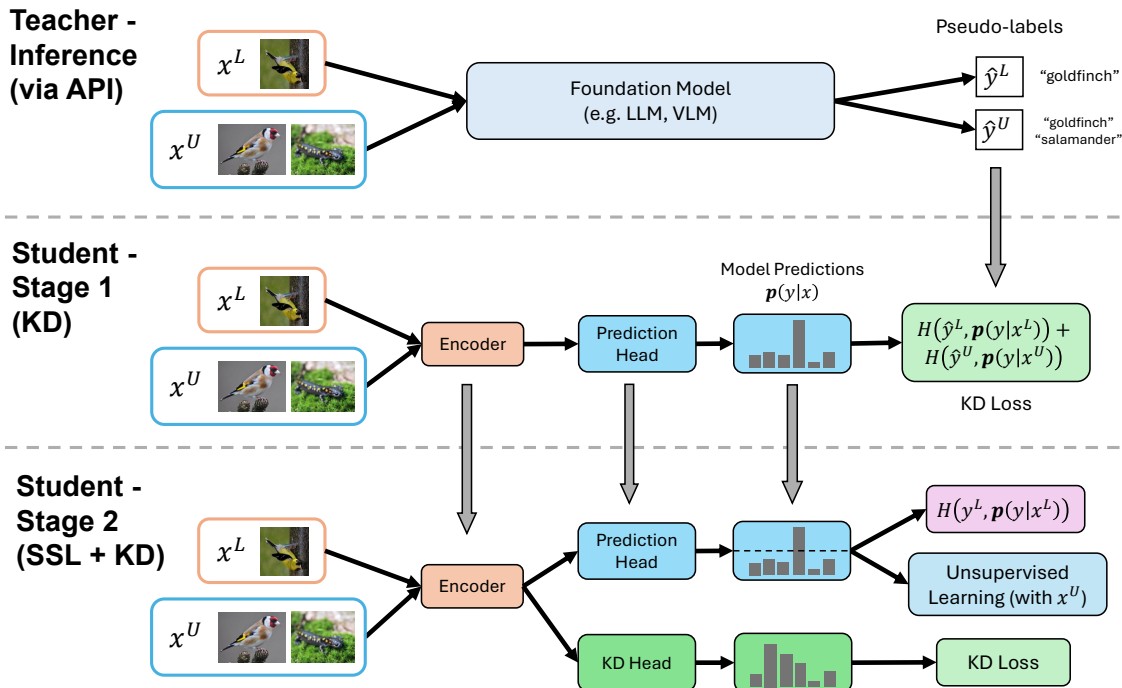

Figure 1: Illustration of the proposed ZeroMatch algorithm. Pseudo-labels from the teacher foundation model are obtained for both labeled and unlabeled input data through an API inference service. In stage 1 of student model training, knowledge distillation is performed with the obtained pseudo-labels. In stage 2, the student model is trained with supervised and unsupervised objective of an SSL algorithm, with weights and initial confident prediction learned from the previous stage. We add an auxiliary classifier head (green box) that runs the same knowledge distillation task from the previous stage, to mitigate potential catastrophic forgetting that may occur during running SSL.

inference services. In such settings, direct fine-tuning of large foundation models (including parameter-efficient fine-tuning (Houlsby et al., 2019; Hu et al., 2022) and test-time adaptation (Sun et al., 2020)) is infeasible due to their large parameter size and high computational costs for backpropagation. Furthermore, naively using pseudo-labels from foundation models to supervise unlabeled data can degrade performance — particularly when pseudo-labels are inaccurate (Zhu et al., 2024) — which can be common when the foundation model has not encountered data similar to the user's personal data. Among existing approaches, knowledge distillation-based methods (Zhao et al., 2023; Vemulapalli et al., 2023; Jiang et al., 2023) offer a promising solution for such resource-constrained settings. However, current methods typically leverage either the teacher's predictions or the labeled data independently, missing opportunities to combine these complementary supervision sources. Thus, fully leveraging labels, unlabeled data, and pseudo-labels from foundation models in a unified framework remains an underexplored direction.

In this paper, we introduce **ZeroMatch**, a method that integrates knowledge distillation and semi-supervised learning to jointly supervise a student model using both pseudo-labels from a foundation model and a confidence-based SSL objective. Our approach is based on the insight that the student model can iteratively refine its predictions on unlabeled data by drawing from the strengths of both the teacher (via distillation) and labeled data (via SSL). We conduct extensive experiments across six datasets in vision and language classification, evaluating ZeroMatch using pseudo-labels generated by both high- and low-quality foundation models. Our results show that ZeroMatch consistently outperforms standard SSL baselines, zero-shot classification, and other zero-shot augmented approaches, highlighting its robustness and practical applicability in low-resource scenarios.

In summary, our key contributions are as follows:

1. We propose ZeroMatch: a novel knowledge distillation-based semi-supervised learning approach that effectively leverages labels, unlabeled data, and teacher pseudo-labels from foundation models.

2. Our framework trains small models while directly leveraging zero-shot predictions from foundation models as pseudo-labels, enabling training on modest hardware, and eliminating the need for the teacher model to be present. This addresses a practical setting where users have limited computational resources but access to foundation model inference services.

3. We provide extensive experiments on six datasets across vision and language domains, which demonstrate that ZeroMatch outperforms both standard SSL methods and zero-shot augmented approaches.

## 2 Related Work

### 2.1 Semi-supervised Learning

Semi-supervised learning (SSL) has evolved from foundational consistency regularization methods to sophisticated deep learning approaches. The Π-Model (Laine & Aila, 2022) and Mean Teacher (Tarvainen & Valpola, 2017) established core principles by enforcing consistent predictions across different model states. FixMatch (Sohn et al., 2020) later unified these ideas by combining weak and strong augmentations with pseudo-labeling. Subsequent work focused on reliability: UPS (Rizve et al., 2021) introduced confidence-based filtering while FlexMatch (Zhang et al., 2021) developed adaptive thresholding for pseudo-label selection. Recent approaches like SimMatch (Zheng et al., 2022) have further advanced the field by incorporating contrastive learning principles.

### 2.2 Foundation Models in SSL

The integration of foundation models into SSL frameworks is an emerging direction. One of the main emerging approaches is to fine-tune foundation models with SSL or unsupervised algorithms (Shi et al., 2023; You et al., 2024; Hegselmann et al., 2023; Menghini et al., 2023; Zhang et al., 2024; Gan & Wei, 2024). These methods provide a strong and robust learner in general, but are not often applied to state-of-the-art foundation models due to reliance on model-specific features like cosine similarity of embeddings, and the significant compute for the fine-tuning step involved in tuning large foundation models. In contrast, our setup leverages foundation models as a pseudo-label generators. Researchers have also proposed theoretical guarantees to accommodate potentially noisy pseudo-labels (Zhu et al., 2024). Our work builds on this goal to develop a practical method for incorporating unreliable pseudo-labels.

### 2.3 Knowledge Distillation in SSL

Knowledge distillation (KD) has been widely used to transfer knowledge from large teacher models to smaller student models (Buciluǎ et al., 2006; Hinton et al., 2015; West et al., 2021; Beyer et al., 2022; Hsieh et al., 2023; Vöge et al., 2024), allowing the benefits of large models on low-compute devices. Many KD approaches applied in the semi-supervised learning (SSL) setting pre-train the teacher model, either via supervised or self-supervised objectives (Chen et al., 2020; He et al., 2021; Yang et al., 2025; Xie et al., 2022; Shi et al.; Vemulapalli et al., 2023). Among them, SRD (Yang et al., 2025) deploys an auxiliary objective to match teacher and student outputs, similar to ours. Methods that rely on teacher model pre-training often make it impractical to apply them to recent large foundation models like Llama-70B due to their large computational requirements for tuning. Recent alternative KD attempts avoid explicitly training a teacher by optimizing the student models on teacher's zero-shot outputs, but trains the student model on either one of teacher model's outputs (Zhao et al., 2023; Jiang et al., 2023) or labeled data (Vemulapalli et al., 2023), missing the synergic opportunity of using both. To the best of our knowledge, there is no existing KD approach that combines SSL-specific features (such as strong augmentation and confidence-based sampling) with label-based KD, which satisfies our goal of fully leveraging labels, unlabeled data, and teacher outputs simultaneously.

## 3 Preliminaries

### 3.1 Problem Setup

We consider an on-device training setup with a compute-limited device (e.g. single GPU, or edge device with memory less than 8GB), semi-supervised training data, and read-only access to a foundation model $\hat{f}$ through an inference service. The training data consists of a labeled set $\mathcal{D}_L = (x_i^L, y_i^L) : i \in [N_L]$ and an unlabeled set $\mathcal{D}_U = (x_i^U) : i \in [N_U]$, where inputs of both sets come from the same distribution. $\mathcal{D} = \mathcal{D}_L \cup \mathcal{D}_U$ represents the full dataset where $[N]$ denotes integers $1, 2, ..., N$. The foundation model $\hat{f}$ generates pseudo-label distributions $\hat{y}_i^L = \hat{f}(x_i^L)$ for labeled data and $\hat{y}_i^U = \hat{f}(x_i^U)$ for unlabeled data. The goal is to train a classifier $f$ that outputs class distributions $\mathbf{p}(y|x)$ using $\mathcal{D}_L$, $\mathcal{D}_U$, and pseudo-labels $\hat{y}_i^L, \hat{y}_i^U$. [2] We assume $\hat{f}$ was not trained on $\mathcal{D}_L$ or $\mathcal{D}_U$. The pseudo-label quality depends on the model's capabilities and prompt design, with potential failure modes including hallucinations and out-of-domain responses.

Our main objective is to develop an SSL method that maximally leverages labels, unlabeled data, and pseudo-labels from the foundation model. This implies that the algorithm should fully utilize pseudo-labels when they are high quality, and gracefully degrade to standard SSL performance when pseudo-labels are unreliable.

### 3.2 Semi-supervised learning methods

Recent advancements in SSL have led to the development of methods that jointly train a classifier from both supervision on the labeled set, and unsupervised feature learning on unlabeled data. The training objective is typically expressed as $\mathcal{L} = \mathcal{L}_s + \mathcal{L}_u$ where $\mathcal{L}_s$ is the supervised loss, often the cross-entropy loss computed on the labeled data:

$$\mathcal{L}_s = \frac{1}{B_L} \sum_{i=1}^{B_L} \mathcal{H}(y_i^L, \mathbf{p}(y|x_i^L)).$$

where $B_L$ is the size of the data batch sampled from the labeled set $\mathcal{D}_L$.

On the other hand, $\mathcal{L}_u$ represents the unsupervised loss, which leverages different strategies to incorporate unlabeled data into the training process. Some notable strategies include the following:

- **Self-training**: This strategy involves using the model trained on labeled data to generate temporary predictions for the unlabeled samples. These predictions are then incorporated into the model's supervision objective (Lee, 2013).

- **Confidence-based sample selection**: Unlabeled samples with less confidence or potentially incorrect pseudo-labels are filtered out based on predefined thresholds. This ensures that only samples with reliable predictions contribute to the training process (Xie et al., 2020).

- **Strong augmentation**: To further enhance feature learning, the model is optimized so that its predictions on weakly augmented and strongly augmented versions of the same sample agree, encouraging the model to learn robust features (Xie et al., 2020).

- **Distribution alignment**: This technique adjusts the output probability distribution of each class based on the input data distribution (Berthelot et al., 2020).

With strong augmentation $\mathcal{A}_s(x_i^U)$ and weak augmentation $\mathcal{A}_w(x_i^U)$ of unlabeled samples $x_i^U$, we denote prediction logits from classifier $f$ of these augmented samples by $p_i^s = \mathbf{p}(y|\mathcal{A}_s(x_i^U))$ and $p_i^w = \mathbf{p}(y|\mathcal{A}_w(x_i^U))$, respectively.

---

[2]Our usage of "pseudo-labels" differs from traditional SSL literature, where they typically refer to predictions on $\mathcal{D}_U$ from a model partially trained on $\mathcal{D}_L$. Here, they represent foundation model predictions without any task-specific training.

When a data batch of size $B_U$ is sampled from $\mathcal{D}_U$, the unsupervised training objective in typical SSL methods takes the following form:

$$\mathcal{L}_u = \frac{1}{B_U} \sum_{i=1}^{B_U} \mathbb{1}(\max(p_i^w) > \tau) \mathcal{H}(\hat{p}_i^w, \mathbf{p}(y|x_i^U))$$

where $\hat{p}_i^w = DA(p_i^w)$ is the label prediction for input $\mathcal{A}_w(x_i^U)$ and $DA$ represents the distribution alignment process.

## 4 The ZeroMatch Framework

In order to develop an SSL method that effectively leverages labeled data, unlabeled data, and pseudo-labels generated by a foundation model, we propose a hybrid approach that combines the strengths of knowledge distillation and SSL techniques.

The two key intuitions behind our approach are:

1. **Knowledge distillation methods utilize the teacher model's predictions** to improve the student model's prediction output on unlabeled data.

2. **Semi-supervised learning algorithms utilize the labeled data** to refine the model's predictions on unlabeled data.

These imply that both methods can be used to improve predictions on unlabeled data from different supervision sources in a complementary manner. To this end, we propose the following two-stage algorithm that integrates both paradigms to maximize the utility of all available supervision signals.

**Stage 1: Knowledge Distillation (KD)** We start by following a standard knowledge distillation procedure with the teacher's output. A student model is trained using available unlabeled data (including the unlabeled input in labeled data) with the pseudo-labels $\hat{y}_i^L, \hat{y}_i^U$ generated from the teacher foundation model $\hat{f}$ with the following objective:

$$\mathcal{L}_{KD} = \frac{1}{N} \left( \sum_{i=1}^{N_L} \mathcal{H}(\hat{y}_i^L, \mathbf{p}(y|x_i^L)) + \sum_{i=1}^{N_U} \mathcal{H}(\hat{y}_i^U, \mathbf{p}(y|x_i^U)) \right).$$

where $N = N_L + N_U$. The resulting student model obtains high-confidence predictions for unlabeled data that match the teacher's pseudo-labels.

**Stage 2: Semi-supervised learning with auxiliary KD loss** The objective of the second stage is to train the student model with an SSL objective. The key component is that, unlike running SSL from scratch, predictions obtained from the previous stage can be utilized as initial high-confidence samples in the SSL algorithm, allowing more unlabeled samples to be utilized from the beginning stage of SSL learning, achieving faster convergence as a result.

One possible failure mode of running an SSL algorithm in stage 2 is catastrophic forgetting (Goodfellow et al., 2013), where learned knowledge from the teacher's pseudo-label can be forgotten during training for the downstream task (SSL in this case). This can happen frequently in a low-label setting, where the SSL algorithm can develop inaccurate predictions on unlabeled data due to inherently limited information in labels, leading to inaccurate pseudo-label that overwrites the knowledge learned. To address this issue, we include knowledge distillation as an auxiliary objective alongside the main SSL objective, motivated by Kar et al. (2022). When the student classifier $f$ consists of a non-linear projector head $h(\cdot)$ and a feature encoder $g(\cdot)$, such that $f = h \circ g$, we introduce an additional linear projector head $h_p(\cdot)$, which learns a knowledge distillation task, while sharing the encoder $g$ with the main classification task. Given a batch of labeled and

unlabeled data of size $B_L$ and $B_U$, stage 2 optimizes $h_p \circ g$ with the following loss:

$$\mathcal{L}_{KD_2} = \frac{1}{B} \left( \sum_{i=1}^{B_L} \mathcal{H}(\hat{y}_i^L, \mathbf{q}(y|x_i^L)) + \sum_{i=1}^{B_U} \mathcal{H}(\hat{y}_i^U, \mathbf{q}(y|x_i^U)) \right).$$

where $B = B_L + B_U$ and $\mathbf{q}(y|x)$ indicates the predicted class distribution of $h_p(g(x))$. Together with the SSL objective, the overall loss function of stage 2 is:

$$\mathcal{L}_{KD-SSL} = \mathcal{L}_s + \mathcal{L}_u + \alpha_t \cdot \lambda_p \mathcal{L}_{KD_2}$$

where $\alpha_t$ is an annealing parameter that linearly increases its value from 0 to 1 during training based on training step $t$, and $\lambda_p$ is a fixed scalar hyperparameter indicating the relative weight of pseudo-label prediction task. We denote the inclusion of annealing with a binary variable $\alpha_p$.

The auxiliary KD head acts as an indirect regularizer that mitigates the catastrophic forgetting issue, by providing continuous supervision of the teacher's pseudo-label to student's model during the SSL training. Because $h_p$ and $h$ share the encoder $g$, the KD loss constrains $g$'s representations to remain aligned with teacher pseudo-labels even as $h$ adapts to the SSL objective. This prevents the encoder from drifting toward erroneous SSL predictions in the early stage of training, which may easily override the teacher knowledge learned from stage 1. $\lambda_p$ and $\alpha_t$ can be set to control the balance between the SSL objective and KD loss to ensure that the KD loss does not overwhelm the SSL objective but is strong enough to keep the stage 1 knowledge. Note that when $\lambda_p$ is set too large, the KD task may dominate the training objective and interfere with the SSL's objective, reducing the flexibility of the SSL head $h$ to adapt to the downstream task. Our algorithm is illustrated in Figure 1.

Our algorithm can robustly improve using both labels and pseudo-labels. When the pseudo-labels are accurate, the model benefits from the knowledge distillation task throughout both stages of training. Conversely, when the pseudo-labels are noisy, the model can still benefit from the SSL's objective, with less impact from the inaccurate pseudo-label due to the KD task being learned at the separate classification head $h_p$. We integrate our method with AdaMatch (Berthelot et al., 2021), which is a highly performant SSL baseline across various datasets and domains, as demonstrated by benchmark results in Wang et al. (2022b). We denote this method by **ZeroMatch**.

While our method is designed to train an accurate student classifier for a given task, note that its encoder may have representation drift due to forcing supervision with teacher pseudo-labels, which may be different from the student's prediction. For applications that requires embeddings from the encoder, we suggest training a separate student model with an additional distillation stage with the current student's output.

## 5   Aren't existing LLM adaptation solutions good enough?

The problem of transferring the knowledge of large language models (LLMs) to domain-specific tasks has been widely studied, and many apporaches are applicable to semi-supervised learning as well. Two prominent strategies among them are: fine-tuning (including its parameter-efficient variants) and prompt engineering. While these approaches have proven effective in many fully supervised settings, they present limitations when applied to semi-supervised learning tasks under low-compute environments. In this section, we highlight these limitations and discuss how ZeroMatch provides a more practical alternative.

**(Parameter-efficient) fine-tuning cannot avoid large memory footprint.**   In terms of statistical performance, fine-tuning open-weight LLMs—including parameter-efficient fine-tuning methods (Hu et al., 2022; Lester et al., 2021) and test-time adaptation techniques (Sun et al., 2020; Hong et al., 2023; Song et al., 2023)—often achieves the strong results, as these methods allow models to fully exploit their rich parameterization. However, such approaches are resource-intensive: both model loading and gradient backpropagation scale with the full model size. We illustrate this limitation in Fig. 2, which compares LoRA fine-tuning and ZeroMatch using the LLaMA-70B model as an example. LoRA requires roughly 140 GB of GPU memory just to load model parameters, and more than 160 GB when including intermediate activations during backpropagation, making it infeasible for users without access to high-end GPU clusters. In contrast,

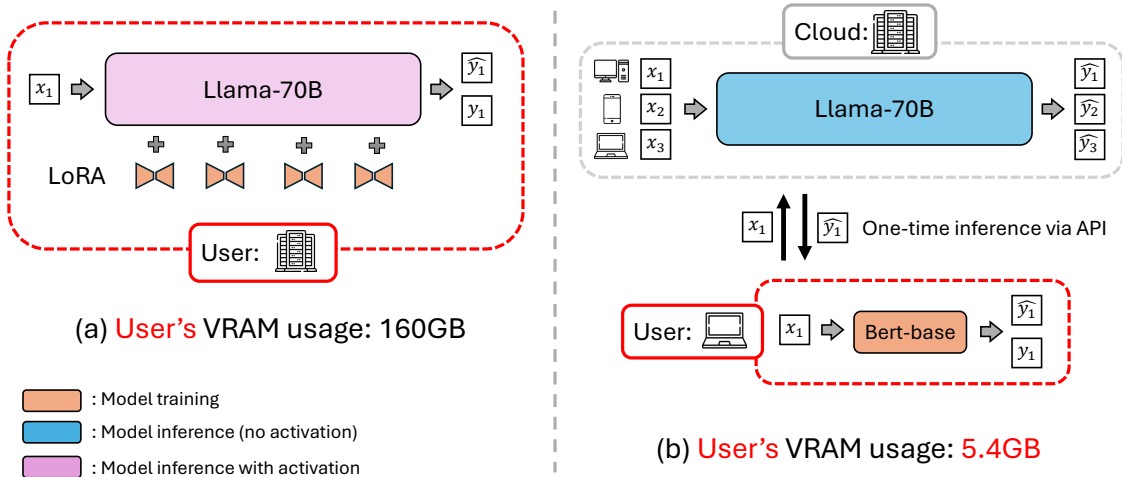

Figure 2: Comparing training setup and memory usage of (a) LoRA fine-tuning and (b) ZeroMatch using the Llama-70B foundation model (with 16 bit quantization). (a) While LoRA offers parameter-efficient adaptation, it still requires loading the entire Llama-70B model and storing intermediate activations for backpropagation, leading to a large memory footprint (approximately 160 GB), requiring users to have access to data center-level GPUs. (b) In contrast, ZeroMatch trains a lightweight student model (e.g., BERT-base) locally while delegating heavy LLM inference to cloud servers via one-time API calls, where cloud servers effectively hide the computation cost by batch processing requests from multiple users. This design offloads both computation and memory from the user, enabling training on low-memory devices (e.g., laptops or smartphones) with only 5.4 GB VRAM usage.

ZeroMatch enables the use of LLMs in an SSL fine-tuning task with a smaller training backbone on modest hardware, such as a single GPU or even edge devices. This decouples the use of high quality information from LLMs from the computational demands of training, allowing effective model training locally without incurring high computational costs.

**Prompt engineering cannot utilize semi-supervised datasets effectively.** Prompt engineering approaches, such as in-context learning, are often used to leverage a few labeled examples by placing them directly within the model's input context. While this strategy enables models to adapt to new tasks without computation-heavy parameter updates, it commonly yields underwhelming performance compared to fine-tuning-based methods (Lehman et al., 2023), particularly when the target task involves complex feature–label relationships that require extensive supervised training. Moreover, prompt-based methods are inherently limited in utilizing large unlabeled datasets due to the restricted context window of foundation models. Even if it becomes technically possible with increased context window size, most foundation models are not trained to interpret or exploit unlabeled data, making effective semi-supervised learning infeasible with this solution.

While in-context learning could, in principle, be employed to generate pseudo-labels for unlabeled data—potentially complementing ZeroMatch's zero-shot responses—we exclude this option in our experimental analysis to isolate and evaluate the effects of labeled supervision within our framework.

In addition to the computational and methodological advantages discussed above, ZeroMatch offers further practical benefits that extend its usability across a wider range of real-world constraints, as described below.

**ZeroMatch is compatible with closed-source LLMs** Many high-performing LLMs (e.g., GPT-4o) are closed-weight and accessible only through limited APIs. These APIs often restrict training capabilities—for instance, OpenAI's fine-tuning API supports only supervised learning with fixed input-output pairs, making training with an SSL algorithm infeasible. ZeroMatch remains compatible in such scenarios. Extracting pseudo-labels from the LLM's text output is straightforward and does not require access to internal representations

Table 1: Summary of data used in the experiments. For each dataset, we run experiments with at most 3 pseudo-label sets from different foundation models. The foundation model $\hat{f}$ used to generate each pseudo-label set are shown below.

| Domain | Dataset | # Train | # Val | #Test | # Class | Foundation model $\hat{f}$ |
|--------|---------|---------|-------|-------|---------|----------------------------|
| NLP | Amazon Review | 250,000 | 25,000 | 650,000 | 5 | GPT-4o, Llama3.3-70B, FLAN-T5 |
| | AG News | 100,000 | 10,000 | 7,600 | 4 | |
| | Yahoo! Answer | 500,000 | 50,000 | 60,000 | 10 | |
| CV | CIFAR100 | 50,000 | - | 10,000 | 100 | GPT-4.1, CLIP |
| | Flowers102 | 1,020 | - | 6,149 | 102 | |
| | Resisc45 | 3,150 | - | 25,200 | 45 | |

or gradients. This makes our approach practical even when constrained to inference-only interactions with proprietary LLMs.

**ZeroMatch reduces data-leakage risks.** Using LLMs often requires uploading data to remote servers where the model is hosted, raising concerns of data leakage—especially when working with privacy-sensitive domains such as healthcare or finance. Assuming that the model training happens on a private device, our setup significantly reduces this risk by limiting data exposure to remote machines to a one-time inference call. Once the data is processed, it can be immediately discarded, eliminating the need for persistent storage or repeated transmission. This is notably safer than fine-tuning LLMs, which requires storing the data on a remote machine for continuous access throughout the training process.

# 6 Experiments

## 6.1 Experiment Setup

**Datasets** We use 6 publicly-available datasets across vision and natural language processing (Table 1). Dataset sizes range from 100s to 100,000s of data points. To evaluate the effect of label size, for each dataset, we create up to three tasks with different sizes of labeled data while the total number of unlabeled data remains the same. When sampling the labeled datapoints, same number of labeled samples are drawn for each class. For example, in the CIFAR100 dataset, we create sets with 100, 200, and 400 labeled data points, each having 1, 2, 4 labels per class, respectively. Detailed explanations of each dataset are provided in Appendix A.1.

**Teacher pseudo-labels from foundation models** For pseudo-labels of each dataset, we consider multiple sets of zero-shot responses with varying quality. For NLP datasets, we generate pseudo-labels with GPT-4o (OpenAI, 2024), Llama3.3-70B (AI, 2024), and FLAN-T5 (Chung et al., 2024) with a standard zero-shot prompt of task description (e.g. *"Select the topic that the given article is about."* or *"What is the sentiment of this review?"*) with the list of the task's label options, and the response is parsed to find the predicted label. Among FLAN-T5 models with different model sizes, we choose two models: one with the best quality, and one having the lowest quality, based on their test accuracy. We name each FLAN-T5 pseudo-label set as 'A' and 'B' in order of decreasing quality. See Appendix A.2.2 for exact models used. For image datasets, we use GPT-4.1 (OpenAI, 2025) and CLIP (Radford et al., 2021). GPT-4.1 is prompted via chat-completion API with the image to select the correct label with standard prompt (e.g. *"Classify the input image into exactly one of the following types."* followed by candidate labels, while CLIP uses its standard zero-shot classification pipeline by comparing image embeddings with text-prompted class embeddings with cosine similarity score. All pseudo-labels are computed once through offline inference without any additional calibration or consistency adjustments. We provide benchmark results of zero-shot pseudo-labels on the test set in Table 12 in Appendix A.2.2. Further details on pseudo-label generation procedures including the exact prompt used are discussed in Appendix A.2.

**Training setup** Data-specific training configurations are explained in detail in Appendix A.3. For image classification, we train ViT-Small (Dosovitskiy, 2020) with AdamW optimizer for $T = 204{,}800$ steps for CIFAR100, and $T = 102{,}400$ steps for Flowers102 and Resisc45. For NLP classification tasks, we use pre-trained BERT-Base (Devlin, 2018) with the AdamW optimizer for $T = 102{,}400$ steps. To ensure a fair comparison with AdaMatch (an SSL baseline), we use the exact same hyperparameters used for AdaMatch. For hyperparmeters specific to our method, we use $\alpha_p = 1$ (indicating that the annealing is applied) and $\lambda_p = 1$ for all experiments. Discussions on the default choice of $\alpha_p$ and $\lambda_p$, and their sensitivity analysis is provided in Appendix 6.5.3.

**Implementation details** We implement our method using the Unified SSL Benchmark (USB) framework (Wang et al., 2022a). To minimize the complexity of integrating our method with existing SSL baselines, we simply add an additional multi-layer perceptron (MLP) classification head with the same architecture as the baseline model's MLP head. All experiments are implemented in PyTorch and run with a single NVIDIA A5000 24GB GPU, but we note that our experiments can run on a GPU with much smaller VRAM due to the training backbone being small (see Section 6.5.4 for computational requirements of our method.).

## 6.2 Baselines

### 6.2.1 Pseudo-supervision

When given a semi-supervised dataset that includes pseudo-labels, a straightforward approach is to fill in the prediction targets for the unlabeled samples using the pseudo-labels, resulting in a fully pseudo-labeled unlabeled set $\hat{\mathcal{D}}_U = \{(x_1^U, \hat{y}_1^U), (x_2^U, \hat{y}_2^U), \ldots, (x_{N_U}^U, \hat{y}_{N_U}^U)\}$. The model can then be trained using a supervised learning algorithm on this fully-labeled dataset $\hat{\mathcal{D}} = \mathcal{D}_L \cup \hat{\mathcal{D}}_U$. We refer to this method as *pseudo-supervision*. Pseudo-supervision has been shown to improve performance in semi-supervised learning scenarios, particularly when a zero-shot method provides accurate pseudo-labels for the task at hand (Hegselmann et al., 2023; Nam et al., 2023). However, when the zero-shot predictions are inaccurate, it remains unclear whether pseudo-supervision will still be beneficial. We refer to Appendix A.4 for implementation details.

### 6.2.2 Pseudo-label as feature input

Another straightforward approach to use pseudo-labels to improve learning is to include them as one-hot features, which is often employed in tabular learning settings to include categorical features (Borisov et al., 2022). For this baseline, we run an SSL algorithm (AdaMatch) with one-hot converted pseudo-labels as an additional feature input to the model. We name this approach *PL feature input*. This approach can learn to accommodate or ignore the pseudo-label features in SSL depending on how helpful they are in learning the task. See Appendix A.4 for implementation details.

### 6.2.3 Additional comparisons with previous works

We discuss additional comparisons of ZeroMatch with other previous works in Appendix B. While our work mainly focuses on involving FM to an SSL setting in a pseudo-label format, there are other previous works outside this category that also try to enhance SSL with FMs, such as fine-tuning FMs for SSL (e.g. GRIP (Menghini et al., 2023), CPL (Zhang et al., 2024) and FineSSL (Gan & Wei, 2024)). We provide an overview of how these methods compares with ZeroMatch in practice in Appendix B.1. We also discuss comparisons with the doubly-robust self-training method (Zhu et al., 2024) which addresses a problem similar to ours in Appendix B.2.

## 6.3 ZeroMatch Enhances SSL with High Quality Pseudo-labels from Large Foundation Models

Benchmark results for the NLP datasets with GPT-4o and Llama3.3-70B are presented in Table 2, and image datasets with GPT-4.1 are presented in Table 3. When combined with pseudo-labels from high-quality large foundation models, ZeroMatch achieves the highest scores in most of the settings (22 of 23 total settings, except 2000-label case of Yahoo Answers with Llama3.3-70B). Our results show that ZeroMatch consistently improves upon SSL baseline and zero-shot, implying effective usage of both labels and zero-shot pseudo-labels

Table 2: Accuracy (%) in Yahoo Answers, AG News, Amazon Review with pseudo-labels from GPT-4o and Llama3.3-70B. Median and standard deviation of 3 different random seeds are reported. Best score among experiments with same pseudo-label set are in bold. 'FM' refers to the foundation model used to generate the pseudo-label set.

| | Dataset | Yahoo Answers | | | AG News | | | Amazon Review | | |
|---|---|---|---|---|---|---|---|---|---|---|
| FM | Label size | 250 | 500 | 2000 | 20 | 40 | 200 | 125 | 250 | 1000 |
| None | Adamatch | 64.81±1.29 | 67.3±0.53 | 69.42±0.42 | 88.13±2.55 | 85.21±1.44 | 88.33±1.94 | 45.03±4.05 | 52.39±1.65 | 56.18±0.45 |
| GPT-4o | Zero-shot | 68.81±0.00 | 68.81±0.00 | 68.81±0.00 | 86.25±0.00 | 86.25±0.00 | 86.25±0.00 | 59.14±0.00 | 59.14±0.00 | 59.14±0.00 |
| | Pseudo-supervise | 67.68±0.06 | 67.30±0.20 | 67.56±0.06 | 86.13±0.15 | 86.33±0.08 | 86.33±0.14 | 56.94±0.27 | 56.65±0.21 | 56.83±0.25 |
| | PL feature Input | 70.56±0.27 | 71.00±0.57 | 71.47±0.67 | 86.25±0.46 | 88.00±0.90 | 88.39±1.00 | 50.38±0.67 | 53.30±1.23 | 57.08±0.39 |
| | ZeroMatch (ours) | **70.90±1.07** | **71.11±0.09** | **72.09±0.32** | **88.50±0.22** | **88.70±0.22** | **88.76±1.20** | **60.12±0.33** | **59.82±0.39** | **60.19±0.48** |
| LLama3.3 -70B | Zero-shot | 69.15±0.00 | 69.15±0.00 | 69.15±0.00 | 88.41±0.00 | 88.41±0.00 | 88.41±0.00 | 55.79±0.00 | 55.79±0.00 | 55.79±0.00 |
| | Pseudo-supervise | 67.85±0.21 | 67.67±0.39 | 67.17±0.14 | 88.47±0.08 | 88.53±0.18 | 88.45±0.08 | 53.74±0.33 | 54.34±0.11 | 54.14±0.53 |
| | PL feature input | 70.70±0.23 | 71.37±0.21 | **72.18±0.25** | 86.79±7.98 | 88.41±0.02 | 88.24±0.92 | 51.55±0.93 | 54.39±0.79 | 57.01±0.37 |
| | ZeroMatch (ours) | **71.28±0.29** | **71.46±0.95** | 71.68±0.19 | **88.79±0.05** | **88.96±0.07** | **88.86±0.17** | **57.36±0.81** | **59.46±0.13** | **60.06±0.28** |

Table 3: Accuracy (%) in CIFAR-100, Flowers102, Resisc45 with pseudo-labels from GPT-4.1. Median and standard deviation of 3 different random seeds are reported. Best score among experiments with same pseudo-label set are in bold. 'FM' refers to the foundation model used to generate the pseudo-label set.

| | Dataset | CIFAR100 | | | Flowers102 | Resisc45 |
|---|---|---|---|---|---|---|
| FM | Label size | 100 | 200 | 400 | 204 | 90 |
| None | Adamatch | 71.43±2.48 | 78.32±0.46 | 84.02±0.74 | 86.71±0.69 | 78.87±0.80 |
| GPT-4.1 | Zero-shot | 83.25±0.00 | 83.25±0.00 | 83.25±0.00 | 88.37±0.00 | 79.28±0.00 |
| | Pseudo-supervise | 84.84±0.03 | 84.93±0.27 | 85.12±0.13 | 85.40±0.25 | 79.59±0.18 |
| | PL feature input | 72.81±1.33 | 82.48±1.87 | 85.78±0.44 | 89.53±1.44 | 82.01±1.09 |
| | ZeroMatch (ours) | **88.01±0.05** | **87.97±0.20** | **88.12±0.14** | **95.17±0.88** | **87.83±0.72** |

as supervision sources. For example, in Amazon Review dataset with 40 labels and pseudo-label set from Llama3.3-70B, ZeroMatch achieves 71.28% compared to zero-shot accuracy of 69.15% and Adamatch 64.81%, yielding an improvement of 2.13% and 6.47% respectively. Our method also outperforms pseudo-supervision and PL feature input baselines, with highest improvement coming from 100-label setting of CIFAR100 with GPT-4.1 where ZeroMatch produces 15.2% improvement compared to PL feature input, and 8.23% improvement in the 90-label setting of Resisc45 with GPT-4.1 for pseudo-supervision. This indicates that ZeroMatch is the most efficient in utilizing all supervision sources among baselines that uses the same data.

### 6.4 ZeroMatch Improves Performance on SSL Baselines Despite Low-Quality Teacher Pseudo-labels

We also present benchmark results with pseudo-labels from FLAN-T5 and CLIP. While FLAN-T5 and CLIP models are generally less powerful than GPT-4 and Llama models, this configuration demonstrates the following practical settings:

- **Unseen task data**: Users may train with personal data that is not seen by the foundation models. Based on training data sources listed in technical reports of FLAN-T5 (Chung et al., 2024) and CLIP (Radford et al., 2021) models, we ensure that these models are not trained with our benchmark dataset. On the other hand, recent models like GPT and Llama do not reveal the data source, therefore we cannot ensure that our benchmark datasets (which are public) are not in the training source of these models.

- **Restrictions in model access**: Users may not be able to access state-of-the-art large foundation models and may be limited to using certain types of less powerful foundation models, due to company restrictions, contracts, or high implementation cost involved in accommodating the new model.

NLP benchmark results with FLAN-T5 are presented in Table 4. Image classification results with CLIP models are presented in Table 5. Our results show that ZeroMatch consistently outperforms or maintains the performance of baselines within one standard deviation all other baselines in most cases (26 of 28 total settings, except CIFAR100 with 400-labels and Flowers102), demonstrating its robustness against low-quality teacher pseudo-labels. For example, for Amazon Review with FLAN-T5 pseudo-label set A, ZeroMatch

Table 4: Accuracy (%) in Yahoo Answers, AG News, Amazon Review with pseudo-label from FLAN-T5 models. 'A' and 'B' in FM column indicates using highest and lowest quality pseudo-label sets (decreasing in order) from FLAN-T5 models. Median and standard deviation of 3 different random seeds are reported. Best score among experiments with same pseudo-label set are in bold.

| | Dataset | Yahoo Answers | | | AG News | | | Amazon Review | | |
|---|---|---|---|---|---|---|---|---|---|---|
| FM | Label size | 250 | 500 | 2000 | 20 | 40 | 200 | 125 | 250 | 1000 |
| None | Adamatch | 64.81±1.29 | 67.3±0.53 | 69.42±0.42 | 88.13±2.55 | 85.21±1.44 | 88.33±1.94 | 45.03±4.05 | 52.39±1.65 | 56.18±0.45 |
| A | Zero-shot | 66.63±0.00 | 66.63±0.00 | 66.63±0.00 | 91.43±0.00 | 91.43±0.00 | 91.43±0.00 | 52.37±0.00 | 52.37±0.00 | 52.37±0.00 |
| | Pseudo-supervise | 65.79±0.18 | 66.31±0.46 | 65.92±0.33 | 91.37±0.07 | 91.22±0.08 | 91.42±0.08 | 51.11±0.27 | 50.86±0.27 | 50.41±0.41 |
| | PL feature input | 67.93±0.80 | 69.6±0.39 | 70.82±0.25 | 91.43±5.51 | **91.43±0.01** | 90.70±0.85 | 50.00±1.01 | 54.56±0.67 | 56.53±0.61 |
| | ZeroMatch (ours) | **69.78±0.94** | **70.51±0.19** | **71.81±0.07** | **91.51±0.12** | 91.42±0.21 | **91.62±0.15** | **58.83±0.51** | **59.69±0.60** | **60.95±0.13** |
| B | Zero-shot | 35.29±0.00 | 35.29±0.00 | 35.29±0.00 | 87.06±0.00 | 87.06±0.00 | 87.06±0.00 | 35.7±0.00 | 35.7±0.00 | 35.7±0.00 |
| | Pseudo-supervise | 36.38±0.51 | 36.29±0.38 | 36.37±0.51 | 88.7±0.16 | 88.51±0.20 | 88.41±0.18 | 36.02±0.04 | 35.97±0.05 | 36.04±0.03 |
| | PL feature input | 65.52±0.48 | 65.88±0.89 | 69.29±0.53 | 84.61±2.08 | 86.45±1.55 | 86.66±1.01 | 48.93±1.89 | 53.76±1.62 | 56.06±0.30 |
| | ZeroMatch (ours) | **67.05±0.76** | **67.13±0.73** | **69.61±0.22** | **90.42±0.19** | **90.24±0.08** | **90.25±0.16** | **50.23±1.35** | **53.78±1.74** | **56.12±0.64** |

Table 5: Accuracy (%) in CIFAR-100, Flowers102, Resisc45 with pseudo-labels from CLIP models. Median and standard deviation of 3 different random seeds are reported. Best score among experiments with same pseudo-label set are in bold. 'FM' refers to the foundation model used to generate the pseudo-label set.

| | Dataset | CIFAR100 | | | Flowers102 | Resisc45 |
|---|---|---|---|---|---|---|
| FM | Label size | 100 | 200 | 400 | 204 | 90 |
| None | Adamatch | 71.43±2.48 | 78.32±0.46 | 84.02±0.74 | 86.71±0.69 | 78.87±0.80 |
| CLIP-Large | Zero-shot | 60.65±0.00 | 60.65±0.00 | 60.65±0.00 | 72.13±0.00 | 60.32±0.00 |
| | Pseudo-supervise | 70.97±0.41 | 71.86±0.63 | 74.05±0.42 | 74.19±0.57 | 63.93±0.21 |
| | PL feature input | 71.21±2.56 | 80.11±0.49 | 84.42±0.22 | **89.51±0.42** | 80.85±1.43 |
| | ZeroMatch (ours) | **84.30±0.71** | **85.77±0.05** | **86.13±0.01** | 88.94±1.66 | **83.91±1.54** |
| CLIP-Base | Zero-shot | 48.07±0.00 | 48.07±0.00 | 48.07±0.00 | 60.17±0.00 | 49.63±0.00 |
| | Pseudo-supervise | 62.19±0.26 | 65.96±0.37 | 71.03±0.23 | 64.09±0.30 | 55.82±0.84 |
| | PL feature input | 71.3±3.26 | 78.85±1.45 | **83.88±0.73** | 87.56±0.29 | 80.85±2.26 |
| | ZeroMatch (ours) | **79.78±0.99** | **80.87±0.62** | 82.69±0.23 | **89.61±2.39** | **82.58±0.85** |

achieves an accuracy of 58.83%, outperforming PL feature input's accuracy of 50.00%. Notably, when pseudo-label quality is low, ZeroMatch consistently improves (or at least maintains the performance within one standard deviation) when the number of labels increases. For example, the scores of Yahoo Answers with FLAN-T5 set B increase from 67.05% to 69.61% as the number of labels increases. This implies that our method utilizes labeled samples to correct and improve when pseudo-label may not be helpful. On the other hand, PL feature input baseline fails to improve in AG News when increasing labels from 40 to 200 with FLAN-T5 A set, where the score decreases from 91.43% to 90.70%. Our method remains largely effective against pseudo-supervision under low-quality pseudo-label settings. For example, in the 250-label setting of Yahoo Answers, pseudo-supervision reaches 36.38% accuracy with pseudo-label set B, which is worse than 67.05% of ZeroMatch by a large margin.

## 6.5 A Deeper Look into ZeroMatch

### 6.5.1 Accomodating embeddings can further improve ZeroMatch

While our problem setup focuses on leveraging pseudo-labels from foundation models, embeddings (from foundation models trained with embedding tasks) may also provide information that is helpful for learning the task. We also provide a way to accommodate these embeddings in our framework to further improve performance on target tasks. Please see Appendix A.4 for implementation details. We provide benchmarks of our method leveraging pseudo-labels of GPT-4.1 and embeddings from the CLIP-large model on Flowers102 and Resisc45 in Table 6. Our result shows that adding embedding to our framework can outperform cases where only pseudo-labels from the same foundation models are included, achieving highest score in both datasets. This indicates that our method can be easily extended to mix and match pseudo-labels with embeddings to further improve the performance.

Table 6: Accuracy (%) in Flowers102, Resisc45 with pseudo-labels and embeddings from CLIP models and GPT-4.1. Median and standard deviation of 3 different random seeds are reported.

| Method | FM inference info used | Flowers102 | Resisc45 |
|---|---|---|---|
| Zero-shot | GPT-4.1 pseudo-label | 88.37±0.00 | 79.28±0.00 |
| Zero-shot | CLIP-large pseudo-label | 72.13±0.00 | 60.32±0.00 |
| ZeroMatch (ours) | GPT-4.1 pseudo-label | 95.17±0.88 | 87.65±0.73 |
| ZeroMatch (ours) | GPT-4.1 pseudo-label + CLIP-large embedding | **97.20±0.41** | **88.52±0.77** |

Table 7: Comparison with ZeroMatch without auxliary loss in Yahoo Answers and Amazon Review with pseudo-labels from GPT-4o and LLama3.3 models. Median and standard deviation of accuracy (%) of 3 different random seeds are reported. Best score among experiments with same pseudo-label set are in bold. 'FM' refers to the foundation model used to generate the pseudo-label set.

| | Dataset | Yahoo Answers | | | Amazon Review | | |
|---|---|---|---|---|---|---|---|
| FM | Label size | 250 | 500 | 2000 | 125 | 250 | 1000 |
| GPT-4o | ZM w.o. aux. | 68.22±0.32 | 69.23±0.53 | 71.45±0.23 | 57.89±0.47 | 59.60±0.33 | 60.16±0.34 |
| | ZeroMatch (ours) | **70.90±1.07** | **71.11±0.09** | **72.09±0.32** | **60.12±0.33** | **59.82±0.39** | **60.19±0.48** |
| LLama3.3-70B | ZM w.o. aux. | 69.30±0.53 | 70.54±0.54 | 71.54±0.31 | 56.59±1.78 | 59.04±0.21 | **60.13±0.08** |
| | ZeroMatch (ours) | **71.28±0.29** | **71.46±0.95** | **71.68±0.19** | **57.36±0.81** | **59.46±0.13** | 60.06±0.28 |

Table 8: Comparison with ZeroMatch Stage 2 only in Yahoo Answers and Amazon Review with pseudo-labels from FLAN-T5 models. Median and standard deviation of accuracy (%) of 3 different random seeds are reported. Best score among experiments with same pseudo-label set are in bold. 'FM' refers to the foundation model used to generate the pseudo-label set.

| | Dataset | Yahoo Answers | | | Amazon Review | | |
|---|---|---|---|---|---|---|---|
| FM | Label size | 250 | 500 | 2000 | 125 | 250 | 1000 |
| A | ZM w.o stage 1 | **70.01±0.63** | 69.76±0.22 | 71.22±0.27 | 55.16±0.30 | 55.75±0.65 | 58.53±0.36 |
| | ZeroMatch (ours) | 69.78±0.94 | **70.51±0.19** | **71.81±0.07** | **58.83±0.51** | **59.69±0.60** | **60.95±0.13** |
| B | ZM w.o stage 1 | 66.6±0.13 | 67.05±0.44 | 69.22±0.53 | 45.53±4.91 | 53.19±1.68 | **56.9±0.63** |
| | ZeroMatch (ours) | **67.05±0.76** | **67.13±0.73** | **69.61±0.22** | **50.23±1.35** | **53.78±1.74** | 56.12±0.64 |

### 6.5.2 Ablation study

Two main components that constructsour method (other than SSL training) are separate knowledge distillation (KD) stage, and learning KD as auxiliary loss for SSL's objective. We validate the effectiveness of each component through the following ablation study.

① *Is auxiliary KD loss necessary?* To validate this, we compare our method with the case without auxiliary KD loss, which essentially runs plain AdaMatch algorithm after running KD in stage 1. We name this approach ZeroMatch without auxiliary loss ('ZM w.o. aux.' in tables) and provide benchmark results in Table 7. Our results show that adding auxiliary KD loss helps, especially in the low-label setting. For example, in the 125-label setting of Amazon Review with GPT-4o, our original method improves plain AdaMatch by 2.22%. Since pseudo-labels inferred from labeled data can become inaccurate in low-label settings, adding additional supervision with teacher pseudo-labels can be particularly helpful in this case.

② *Does having separate KD stage necessary?* Since stage 2 of our algorithm also includes auxiliary KD loss, one may wonder if running stage 2 without stage 1 in our algorithm is enough to achieve the goal of merging benefits of KD and SSL. To clarify this, we compare our method with the case without running stage 1 knowledge distillation. We name this approach ZeroMatch without stage 1 ('ZM w.o stage 1' in tables) and provide benchmark results in Table 8. Our results show that while ZeroMatch without stage 1 can sometimes achieve similar performance to original ZeroMatch (ex. in Yahoo Answers) it may also produce a large gap in performance depending on the dataset. For example, in the 250-label setting of Amazon Review with 'A' pseudo-label set, ZM stage 2 only reveals a 3.94% accuracy drop compared to original.

Table 9: Hyperparameter sensitivity results in Yahoo Answers (250 labels) and Amazon Review (125 labels) with pseudo-labels from FLAN-T5 models.

| $\alpha, \lambda$ | Yahoo Answers | Amazon Review |
|---|---|---|
| $(0, 0.5)$ | 69.15±0.46 | 59.55±1.69 |
| $(0, 1.0)$ | 69.90±0.87 | **60.20±1.02** |
| $(1, 0.5)$ | 68.26±0.94 | 59.31±1.43 |
| $(1, 1.0)$ | **70.71±0.36** | 59.47±1.26 |

Table 10: Computational cost on training with CIFAR100.

| Method | VRAM(GB) | Training time (sec) |
|---|---|---|
| Adamatch | 4.2 | 41100 |
| ZeroMatch | 5.4 | 61662 |

Table 11: ZeroMatch's performance combined with 3 different SSL baselines (FixMatch, FreeMatch, SoftMatch) in Yahoo Answers with 4 different FM pseudo-label sets.

| | SSL algo. | FixMatch | | | FreeMatch | | | SoftMatch | | |
|---|---|---|---|---|---|---|---|---|---|---|
| FM | Label size | 250 | 500 | 2000 | 250 | 500 | 2000 | 250 | 500 | 2000 |
| None | SSL only | 54.04±0.53 | 63.56±2.03 | 66.59±0.48 | 60.84±2.91 | 66.27±0.45 | 67.29±0.77 | 62.30±0.91 | 64.84±0.94 | 67.43±0.87 |
| GPT-4o | Zero-shot | 68.81±0.00 | 68.81±0.00 | 68.81±0.00 | 68.81±0.00 | 68.81±0.00 | 68.81±0.00 | 68.81±0.00 | 68.81±0.00 | 68.81±0.00 |
| | ZeroMatch (ours) | **70.43±0.13** | **70.96±0.33** | **72.11±0.22** | **70.91±0.37** | **70.72±0.11** | **72.52±0.28** | **71.46±0.90** | **71.20±0.20** | **72.22±0.25** |
| LLama3.3 -70b | Zero-shot | 69.15±0.00 | 69.15±0.00 | 69.15±0.00 | 69.15±0.00 | 69.15±0.00 | 69.15±0.00 | 69.15±0.00 | 69.15±0.00 | 69.15±0.00 |
| | ZeroMatch (ours) | **70.45±0.59** | **70.75±0.34** | **71.96±0.55** | **70.98±0.43** | **71.17±0.67** | **71.68±0.12** | **71.30±0.23** | **70.93±0.45** | **71.72±0.35** |
| A | Zero-shot | 66.63±0.00 | 66.63±0.00 | 66.63±0.00 | 66.63±0.00 | 66.63±0.00 | 66.63±0.00 | 66.63±0.00 | 66.63±0.00 | 66.63±0.00 |
| | ZeroMatch (ours) | **69.64±0.72** | **71.19±0.20** | **71.78±0.20** | **70.36±0.40** | **70.56±0.83** | **71.47±0.27** | **69.92±0.75** | **70.88±0.64** | **72.05±0.43** |
| B | Zero-shot | 35.29±0.00 | 35.29±0.00 | 35.29±0.00 | 35.29±0.00 | 35.29±0.00 | 35.29±0.00 | 35.29±0.00 | 35.29±0.00 | 35.29±0.00 |
| | ZeroMatch (ours) | **66.65±1.49** | **66.82±0.40** | **69.50±0.26** | **67.79±0.82** | **68.24±0.33** | **70.15±0.10** | **67.64±0.57** | **67.47±0.52** | **69.93±0.23** |

### 6.5.3 Hyperparameter sensitivity analysis

For simplicity of settings, we mainly benchmark our method with hyperparameters $\alpha_p = 1$ and $\lambda_p = 1$ without finding the optimal setting. The reasoning behind this choice is to make the KD loss the same as fully-supervised training when pseudo-labels are accurate, and to enable annealing to allow more freedom to accommodate labeled data in the early stage of AdaMatch, which develops initially confident predictions for unlabeled data.

In this section, we run a sensitivity analysis on combinations of these parameters. We conduct experiments with four different sets: $(\alpha_p, \lambda_p) = (0, 0.5), (0, 1.0), (1, 0.5), (1, 1.0)$. The results with NLP datasets with FLAN-T5 models are shown in Table 9. We find that while $(\alpha_p, \lambda_p) = (1, 1.0)$ can be a good candidate on some datasets (ex. Yahoo Answers), there may exist other optimal parameters that can further improve performance.

### 6.5.4 Computational cost analysis

We provide memory usage and computation cost associated with our method and the SSL baseline(AdaMatch) for CIFAR100 dataset in Table 10. The usages are measured on our machine with 1 NVIDIA A10G GPU. We note that our method consumes around 50% more training time compared with AdaMatch. This is due to the separate knowledge distillation at stage 1, which utilizes half of the batch size of AdaMatch. Additionally, our method uses slightly more VRAM due to back-propagating through weak-augmented samples in KD auxiliary loss in stage 2, while in AdaMatch weak-augmented samples are only used with a forward pass to get predictions and back-propagation happens only in strong augmented samples. Possible strategies to reduce training runtime and memory usage are discussed in Appendix C.1 and Appendix C.2.

### 6.5.5 ZeroMatch with other SSL algorithms

While we mainly use AdaMatch as the SSL baseline to use with ZeroMatch , we also provide benchmarks of our method with 3 additional SSL algorithms: FixMatch (Sohn et al., 2020), FreeMatch (Wang et al., 2022c), and SoftMatch (Chen et al., 2023). Results on the Yahoo Answers dataset with 4 different FMs (GPT-4o,

Llama3.3-70b, FLAN-T5 (A), FLAN-T5 (B)) are shown in Table 11. The results show that ZeroMatch improves both SSL and zero-shot accuracy in all cases, implying ZeroMatch's applicability to wide range of SSL methods. While we expect that ZeroMatch can work with many types of SSL algorithms with confidence-based sampling, we note that counterexamples may exist. One such example is UDA (Xie et al., 2020), which uses training signal annealing (TSA) to utilize only the low-confidence labeled samples in the initial training stage. This procedure conflicts with our solution since we have high-confidence samples at the start of SSL training, resulting in labeled samples not being utilized.

## 7 Conclusion

This work introduces ZeroMatch, a framework for robustly integrating foundation model predictions into semi-supervised learning. Through extensive experiments across multiple datasets from different domains, we demonstrate that ZeroMatch achieves state-of-the-art performance while maintaining robustness to varying pseudo-label quality. Our learning-based mechanism effectively balances between limited labels and plentiful pseudo-labels, enabling practitioners to leverage foundation models without the computational overhead of direct fine-tuning. Future work could explore extending ZeroMatch to more complex tasks and investigating theoretical guarantees for the weighting mechanism.

### Broader Impact

Our work aims to make semi-supervised learning more accessible by reducing the computational resources needed to leverage foundation models. This has positive implications for democratizing AI development, particularly benefiting researchers and organizations with limited computing infrastructure.

However, improved semi-supervised learning techniques could also lower the barrier for developing AI systems with potential misuse cases. While our method focuses on improving efficiency rather than expanding capabilities, we acknowledge the need for careful consideration of deployment contexts.

The adaptive weighting mechanism we introduce could potentially mitigate some biases in foundation models by automatically reducing reliance on unreliable predictions. However, when the number of labeled data are low, this may not hold, and our method may propagate or amplify model biases into downstream student models. Further investigation is needed regarding the method's behavior across different demographic groups and task domains.

## 8 Acknowledgements

This work used the Savio computational cluster resource provided by the Berkeley Research Computing program at the University of California, Berkeley (supported by the UC Berkeley Chancellor, Vice Chancellor for Research, and Chief Information Officer). This work also used Delta GPU at NCSA through allocation CIS250269 from the Advanced Cyberinfrastructure Coordination Ecosystem: Services & Support (ACCESS) program, which is supported by U.S. National Science Foundation grants #2138259, #2138286, #2138307, #2137603, and #2138296.

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

# A   Experiment Details

## A.1   Datasets

We use six datasets from the NLP and image classification domains. We provide a description of each of them.

**CIFAR-100**   CIFAR-100 (Krizhevsky, 2009) is a 32×32 pixel natural image recognition dataset consisting of 100 classes. There are 500 training samples and 100 test samples per class.

**Flowers102**   Flowers102 (Nilsback & Zisserman, 2008) is an image dataset containing 102 flower categories commonly found in the United Kingdom. The dataset contains a minimum of 40 and a maximum of 258 images per class.

**Resisc45**   RESISC45 (Cheng et al., 2017) is a publicly available benchmark for Remote Sensing Image Scene Classification. Images from 45 kinds of scenes are provided.

**Yahoo Answer**   Yahoo Answers (Chang et al., 2008) is a question text topic classification dataset with 10 topic categories. Each class contains 140,000 training samples and 6,000 test samples. Training and validation sets are created by sampling 50,000 and 5,000 samples per class from the training set, and the test set is unchanged, following the data settings in USB (Wang et al., 2022a).

**AG News**   The AG News (Zhang et al., 2015) dataset is a news topic classification dataset with 4 classes. The original dataset contains 30,000 training samples and 1,900 test samples per class. In our experiments, 25,000 samples and 2,500 samples per class are sampled from the training set, following USB (Wang et al., 2022a). The test dataset is unchanged.

**Amazon Review**   The Amazon Review (McAuley & Leskovec, 2013) dataset is a sentiment classification dataset with product review text input. There are 5 classes indicating the review score. Each class contains 600,000 training samples and 130,000 test samples. Following USB (Wang et al., 2022a), 50,000 samples and 5,000 samples per class from the training set are used as the training and validation datasets respectively. The test dataset is unchanged.

We use the data splits provided by USB (Wang et al., 2022b) for the NLP datasets, and the splits provided by CPL (Zhang et al., 2024) for Flowers102 and Resisc45.

## A.2   Obtaining pseudo-labels from foundation models

In this section, we discuss details of obtaining the zero-shot pseudo-labels used to train with ZeroMatch.

### A.2.1   Details of pseudo-label generation process

**Image classification datasets**   For the image classification datasets, we mainly use GPT-4.1 and CLIP models to generate pseudo-labels.

For GPT-4.1 (OpenAI, 2025), we use the chat completion feature to prompt for image classification. We use the following system prompt to describe the task:

> You are an image classification system, asked to classify *{topic}*.
> Classify the input image into exactly one of the following types: *{candidate labels}*
> Respond with only the type that best fits the image.

The topic variable is set as 'flowers' for Flowers102, 'scene' for Resisc45, and 'tiny image' for CIFAR100. Candidate labels are set as the possible labels in text format. The same procedure is applied when using other chat-completion-based models (GPT-4o (OpenAI, 2024), Qwen2-VL (Wang et al., 2024), Llama3.2-Vision (AI,

2024)). After obtaining the output, we parse the output text to check if text labels in the candidate label set exist in the output. In cases where no label is detected, we use a default pseudo-label: `rose` for Flowers102, `commercial area` for Resisc45, and `road` for CIFAR100.

When using the CLIP model, we follow the procedures of zero-shot classification in Radford et al. (2021). We compute the feature embedding of a given image, and for each label in the candidate labels we prompt the model with the following template: `This is a photo of {label}` and compute its embedding. We then obtain the pseudo-label by finding the label that gives the closest embedding to the image embedding based on cosine similarity.

**NLP classification datasets — topic classification: AG News, Yahoo Answers**   To obtain pseudo-labels for topic classification tasks, we use the following prompt template with GPT-4o, Llama3.3-70B, and FLAN-T5 models for a given input text:

> Select the topic that the given article is about. The topics are: *{candidate_labels}*.
>
> Article: *{input_text}*,
> Answer:

For AG News, the candidate labels are `world, sports, business, technology`.  For Yahoo Answers, the candidate labels are `society, science, health, education, computer, sports, business, entertainment, relationship, politics`. If the output does not contain labels from the candidate label set or is not parsable, we set the pseudo-label to be `world` for AG News and `society` for Yahoo Answers as the default label. For GPT-4o and Llama models, we use the chat completion template with the following system prompt: `You are a helpful assistant.`

**NLP classification datasets — sentiment classification: Amazon Review**   We use the following prompt template with GPT-4o, Llama3.3-70B, and FLAN-T5 models for a given input text to obtain the sentiment of the review:

> *{input_text}*
> What is the sentiment of this review?
>
> OPTIONS:
> - very negative
> - negative
> - neutral
> - positive
> - very positive

If the model output is not recognized as one of the candidate labels, we select `neutral` as the default pseudo-label. For GPT-4o and LLaMA models, we use the chat completion template with the following system prompt: `You are a helpful assistant.`

### A.2.2   Zero-shot benchmark results and foundation model selection

To select foundation models to be used with ZeroMatch, we consider (1) the best-performing large foundation models that can provide maximum statistical performance with our method and (2) the lowest-quality foundation models to test the robustness of our method against inaccurate pseudo-labels.

We provide zero-shot benchmark results on the test set in Table 12 (NLP) and Table 13 (Image).

We found that GPT-4o and Llama3.3-70B were the highest performing overall, scoring top-2 on all datasets. Among FLAN-T5 models, we find that FLAN-T5-XXL performs the best on Yahoo Answers and AG News, while the FLAN-T5-XL model performs better on Amazon Review. To benchmark our method with highest and lowest accuracy, FLAN-T5-XXL and FLAN-T5-small were selected as A and B sets of pseudo-labels for

Table 12: Zero-shot pseudo-label accuracies(%) for NLP datasets (test set).

| FM | Yahoo Answers | AG News | Amazon Review |
|---|---|---|---|
| GPT-4o | 68.81 | 86.25 | **59.14** |
| Llama3.3-70B | **69.15** | **88.41** | 55.79 |
| FLAN-T5-XXL | **66.62** | **91.43** | 45.61 |
| FLAN-T5-XL | 63.97 | 91.39 | **52.37** |
| FLAN-T5-large | 61.33 | 87.66 | 51.96 |
| FLAN-T5-base | 55.17 | 88.68 | 42.23 |
| FLAN-T5-small | 29.44 | 87.07 | 35.7 |

AG News and Yahoo Answers, and for Amazon Review, we use FLAN-T5-XL and FLAN-T5-small as A and B sets.

Table 13: Zero-shot pseudo-label accuracies(%) for image datasets (test set).

| FM | CIFAR100 | Flowers102 | Resisc45 |
|---|---|---|---|
| GPT-4.1 | **83.25** | **88.37** | **79.28** |
| GPT-4o | 81.5 | 86.0 | 73.46 |
| Qwen2-VL | 62.58 | 78.11 | 68.25 |
| Llama3.2-90B-Vision | 36.05 | 59.02 | 44.58 |
| CLIP-large-patch14 | 62.27 | 72.13 | 60.32 |
| CLIP-base-patch32 | 49.49 | 60.17 | 49.63 |

For the image classification task, we benchmarked GPT-4.1, GPT-4o, Qwen2-VL, and LLaMA3.2-90B-Vision with our target datasets, and found that GPT-4.1 was best in all tasks (see Table 13). As a result, we use GPT-4.1, CLIP-large, and CLIP-base models for our benchmark.

### A.3 Training setup

**Setup for image classification** We use ViT (Dosovitskiy, 2020) as the backbone of our model and employ pre-trained ViT-Small models provided by the USB framework (Wang et al., 2022a). For CIFAR100, ViT-Small with a patch size of 2 and image size of 32 is used, and for Flowers102 and Resisc45 datasets, ViT-Small with a patch size of 16 and image size of 224 is used. Optimization is performed using the AdamW optimizer with a cosine learning rate schedule given by $\eta_t = \eta_0 \cos\left(\frac{7\pi t}{16T}\right)$, where the initial learning rate is set to $\eta_0 = 5 \times 10^{-4}$ for CIFAR100 and $\eta_0 = 1 \times 10^{-4}$ for Flowers102 and Resisc45. For CIFAR100, we train the model for a total of $T = 204,800$ steps, with a warm-up phase of 5,120 steps, and for Flowers102 and Resisc45, we run for half the number of training steps, $T = 102,400$, with 2,560 warm-up steps to reflect the smaller training set size. The batch sizes for both labeled and unlabeled data are set to 16. For AdaMatch-specific parameters, the threshold for the utilization cutoff mask is set to 0.95. For data augmentation, we apply random cropping and random horizontal flipping for weak augmentation, while RandAugment (Cubuk et al., 2020) is utilized for strong augmentation. For the knowledge distillation stage (stage 1 of ZeroMatch), we run supervised training with teacher pseudo-labels for the same number of training steps. Regarding the hyperparameters specific to our method, we report results with $\lambda_p = 1$ and with annealing ($\alpha_p = 1$). Detailed hyperparameter settings are provided in Table 14.

**Setup for NLP classification** For NLP tasks, we fine-tune the pre-trained BERT-Base (Devlin, 2018) using the AdamW optimizer with a cosine learning rate schedule. The total number of training steps is set to 102,400, with a warm-up phase of 5,120 steps. Both the labeled and unlabeled batch sizes are set to 4. All input text is truncated to ensure its length remains within the context length of BERT-Base. For data augmentation, we employ back-translation (Zhu et al., 2020) using German–English and Russian–English translation as strong augmentations. No weak augmentation is applied, and the original text input is used instead. We run AdaMatch with a cutoff threshold of $\tau = 0.95$. Dataset-specific hyperparameters are detailed in Table 15. For the knowledge distillation stage (stage 1 of ZeroMatch), we run supervised training with

teacher pseudo-labels for the same number of training steps. We report the performance of ZeroMatch with $\lambda_p = 1$ and with annealing ($\alpha_p = 1$).

Table 14: Hyperparameters of Image classification datasets.

| Dataset | CIFAR100 | Flowers102 | Resisc45 |
|---|---|---|---|
| Image Size | 32 | 224 | 224 |
| Model | ViT-S-P4-32 | ViT-S-P16-224 | ViT-S-P16-224 |
| Weight Decay | | 5e-4 | |
| Labeled Batch size | | 16 | |
| Unlabeled Batch size | | 16 | |
| Learning Rate | 5e-4 | 1e-4 | 1e-4 |
| Layer Decay Rate | 0.5 | 0.65 | 0.65 |
| Scheduler | | $\eta = \eta_0 \cos\left(\frac{7\pi k}{16K}\right)$ | |
| Model EMA Momentum | | 0.0 | |
| Prediction EMA Momentum | | 0.999 | |
| Weak Augmentation | Random Crop, Random Horizontal Flip | Random Resized Crop and Interpolation, Random Horizontal Flip | |
| Strong Augmentation | | RandAugment (Cubuk et al., 2020) | |
| $\alpha_p$ | | 1 | |
| $\lambda_p$ | | 1.0 | |

Table 15: Hyperparameters of NLP tasks.

| Dataset | AG News | Yahoo! Answer | Amazon Review |
|---|---|---|---|
| Max Length | | 512 | |
| Model | | Bert-Base | |
| Weight Decay | | 1e-4 | |
| Labeled Batch size | | 4 | |
| Unlabeled Batch size | | 4 | |
| Learning Rate | 5e-5 | 1e-4 | 5e-5 |
| Layer Decay Rate | 0.65 | 0.65 | 0.75 |
| Scheduler | | $\eta = \eta_0 \cos\left(\frac{7\pi k}{16K}\right)$ | |
| Model EMA Momentum | | 0.0 | |
| Prediction EMA Momentum | | 0.999 | |
| Weak Augmentation | | None | |
| Strong Augmentation | | Back-Translation (Xie et al., 2020) | |
| $\alpha_p$ | | 1 | |
| $\lambda_p$ | | 1.0 | |

### A.4 Additional implementation details

**Pseudo-supervision** For the pseudo-supervision baseline, a supervised learning algorithm is run with fully labeled data (filled with FM pseudo-labels). To ensure a fair comparison with SSL baselines, we use the same number of labeled and unlabeled batches participating during optimization. In each optimization step, when a labeled batch of size $B_L$ and an unlabeled batch of size $B_U$ are given, pseudo-supervision optimizes the following loss:

$$\mathcal{L}_{PS} = \frac{1}{B}\left(\sum_{i=1}^{B_L} \mathcal{H}(y_i^L, \mathbf{p}(y|x_i^L)) + \sum_{i=1}^{B_U} \mathcal{H}(\hat{y}_i^U, \mathbf{p}(y|x_i^U))\right),$$

where $B = B_L + B_U$.

**Pseudo-label as feature input** To implement the pseudo-label-as-feature-input baseline, we change the model architecture to accommodate one-hot pseudo-labels. When a training sample $x_i$ with its corresponding one-hot converted pseudo-label $\tilde{y}_i$ passes through the model with encoder $g$ and classifier head $h$, the model concatenates encoder output $g(x_i)$ and the one-hot pseudo-label $\tilde{y}_i$ as additional features, producing $h(g(x_i), \tilde{y}_i)$ as output. Note that the input dimension of head $h$ is adjusted to accommodate the additional feature dimension (number of classes). With each sample paired with a pseudo-label, this modification allows running the SSL baseline (AdaMatch) without further modification, and the model can learn to accommodate or ignore the pseudo-label feature depending on how helpful it is for learning the task.

**Accommodating embeddings to ZeroMatch** To accommodate feature embeddings from foundation models in ZeroMatch, we concatenate embeddings with our encoder output before passing them to the classifier head. When the model has an encoder $g$ and classifier head $h$, we modify the input dimension of the MLP head so that encoded features and embeddings are concatenated and fed into $h$. As a result, when a training sample $x_i$ with its embedding $e_i$ is given, the model outputs $h(g(x_i), e_i)$. The same procedure is applied to the auxiliary KD classifier head for stage 2 in our algorithm, producing $h_p(g(x_i), e_i)$ to be supervised with teacher pseudo-labels.

# B    Additional comparsion with previous works

## B.1    Compare with fine-tuning foundations models for semi-supervised learning

As discussed in Sec. 5, direct fine-tuning large foundation models (or using their parameter-efficient variants) may offer higher statistical performance than ZeroMatch by leveraging their vast number of parameters. However, such approaches incur significantly higher memory and computational costs. In contrast, our setup trains a compact backbone model that requires only modest resources (less than 8GB of GPU memory), making it suitable for resource-constrained environments. For fair comparison, we compare our method against recent approaches that fine-tune FMs for SSL and have computational requirements comparable to ours.

We compare our approach with GRIP (Menghini et al., 2023), CPL (Zhang et al., 2024) and FineSSL (Gan & Wei, 2024), which adopt prompt-tuning strategies for vision-language models such as CLIP (Radford et al., 2021) and iteratively refine pseudo-label predictions based on confidence and features.

While direct comparison of these works with ours is not possible due to settings that differ largely in model and training algorithms, we present scenarios where our method can outperform these approaches, and provide comparison details on training and information used. We benchmark our method on the exact same data settings that these works have conducted their experiments on.

**Visual prompt tuning methods** Visual Prompt Tuning (VPT) (Jia et al., 2022) is a prompt-tuning method for vision-language models that tunes the prefix part of the input layer while keeping the textual encoder fixed. FineSSL, CPL and GRIP provide VPT benchmarks with CLIP models. We provide comparisons of ZeroMatch on FineSSL's CIFAR-100 results in Table 16, and results of GRIP and CPL with Flowers102 and Resisc45 in Table 17.

Table 16: Comparison with FineSSL's VPT (visual prompt tuning) results on CIFAR100 with 400 labels (4 labels per class). * indicates numbers published in FineSSL.

| Method | FM Inference info used | Training backbone | Fine-tuning method | Accuracy |
|--------|------------------------|-------------------|--------------------|----------|
| FineSSL* | - | CLIP ViT-B-16 | VPT-deep | 80.44±0.24 |
| Zero-shot | CLIP ViT-L-14 - pseudo-label | - | - | 60.65±0.00 |
| ZeroMatch (ours) | CLIP ViT-L-14 - pseudo-label | ViT-S-2 | Full fine-tuning | **86.73±1.42** |

Our results show ZeroMatch outperforming all the baseline methods. This is largely due to leveraging high-quality outputs of larger foundation models: for CIFAR-100, we use pseudo-labels from CLIP ViT-L-14,

Table 17: Comparison with GRIP and CPL's VPT results on Flowers102 and Resisc45 datasets with 2 labels per class. * indicates numbers published in the paper.

| Method | FM Inference info used | Training backbone | Finetuning method | Flowers102 | RESISC |
|---|---|---|---|---|---|
| Zero-shot* | CLIP ViT-B-32 - pseudo-label | - | - | 63.67 | 54.48 |
| GRIP* | CLIP text encoder - embedding | CLIP ViT-B-32 | VPT | 67.95±1.2 | 71.22±0.77 |
| CPL* | CLIP text encoder - embedding | CLIP ViT-B-32 | VPT | 73.52±0.37 | 78.46±0.74 |
| Zero-shot | GPT-4.1 pseudo-label | - | - | 88.37 | 79.28 |
| ZeroMatch (ours) | GPT-4.1 pseudo-label | ViT-S-16 | Full fine-tuning | **95.17±0.88** | **87.65±0.73** |

and for Flowers102, we leverage GPT-4.1. We also note that the training backbone we use is smaller than the baselines. We use ViT-Small, which uses fewer attention heads compared to CLIP-ViT-B models.

This result implies that our method can outperform many baselines that fine-tune foundation models by simply leveraging the highest-quality foundation model, while training a smaller model for the task on a low-compute device.

**Textual prompt tuning methods** Textual Prompt Tuning (Zhou et al., 2022) is another common prompt-tuning method which tunes the input prefix of the textual encoder of vision-language models while keeping the visual encoder fixed. The textual encoder is trained with embeddings obtained from a large visual encoder, and it can be implemented using the visual encoder in inference mode only. This setting resembles our problem setup in that the inference outputs of larger models are used to train a smaller model. We provide benchmarks comparing our method to GRIP and CPL's textual prompt tuning results in Table 18.

Table 18: Comparison with GRIP and CPL's TPT (Textual Prompt Tuning) results on Flowers102 and Resisc45 datasets with 2 labels per class. * indicates numbers published in the paper.

| Method | FM Inference info used | Training backbone | Finetuning method | Flowers102 | RESISC |
|---|---|---|---|---|---|
| Zero-shot* | CLIP ViT-B-32 - pseudo-label | - | - | 63.67 | 54.48 |
| GRIP* | CLIP ViT-B-32 - embedding | CLIP text encoder | Textual prompt tuning | 83.6±0.68 | 74.11±0.68 |
| CPL* | CLIP ViT-B-32 - embedding | CLIP text encoder | Textual prompt tuning | 89.66±0.36 | 80.98±0.11 |
| Zero-shot* | CLIP ViT-L-14 - pseudo-label | - | - | 73.98 | 62.67 |
| CPL* | CLIP ViT-L-14 - embedding | CLIP text encoder | Textual prompt tuning | 96.80±0.63 | 87.75±0.29 |
| Zero-shot | GPT-4.1 pseudo-label | - | - | 88.37 | 79.28 |
| ZeroMatch (ours) | GPT-4.1 pseudo-label | ViT-S-16 | Full fine-tuning | 95.17±0.88 | 87.65±0.73 |
| ZeroMatch (ours) | GPT-4.1 pseudo-label + CLIP ViT-L-14 - embedding | ViT-S-16 | Full fine-tuning | **97.20±0.41** | **88.52±0.77** |

While our method paired with pseudo-labels from GPT-4.1 can outperform CPL and GRIP with CLIP-ViT-B, CPL's results with CLIP ViT-L outperform our score. We additionally benchmark our method using embeddings from the CLIP ViT-L model, which CPL uses as the supervision source. The results show that our method improves with added embedding information, outperforming other baselines, implying that embeddings are a helpful information source for learning the benchmark task.

## B.2 Comparison with doubly-robust self-training

Doubly-robust self-training (Zhu et al., 2024) addresses a problem setting similar to ours, proposing a method that robustly improves performance despite varying quality of pseudo-labels. The method employs the following loss function:

$$\mathcal{L}_{DR} = \frac{1}{N_L} \sum_{i=1}^{N_L} \mathcal{H}(y_i^L, \mathbf{p}(y|x_i^L)) - \frac{1}{N_L} \sum_{i=1}^{N_L} \mathcal{H}(\hat{y}_i^L, \mathbf{p}(y|x_i^L)) + \frac{1}{N} \left( \sum_{i=1}^{N_L} \mathcal{H}(\hat{y}_i^L, \mathbf{p}(y|x_i^L)) + \sum_{i=1}^{N_U} \mathcal{H}(\hat{y}_i^U, \mathbf{p}(y|x_i^U)) \right).$$

When the pseudo-labels are accurate, the loss function approximates training with the full dataset $\mathcal{D}_L$ and $\mathcal{D}_U$, both with ground truth labels. Conversely, when the pseudo-labels are inaccurate, the loss function

asymptotically approaches training with only the labeled samples $\mathcal{D}_L$. The work provides convergence improvement guarantees over training with labeled data alone, under varying pseudo-label quality, under certain regularity conditions. While the study presents experimental results using a single pseudo-label set inferred from a pre-trained model, its practical effectiveness with inaccurate pseudo-labels in low-label settings is unexplored. To implement this method and ensure a fair comparison with ours, we use the same number of labeled and unlabeled batches participating during the optimization process. In each optimization step, when a labeled batch of size $B_L$ and an unlabeled batch of size $B_U$ are given, the method optimizes the following:

$$\mathcal{L}_{DR} = \frac{\alpha_t}{B_L} \sum_{i=1}^{B_L} \mathcal{H}(y_i^L, \mathbf{p}(y|x_i^L)) - \frac{\alpha_t}{B_L} \sum_{i=1}^{B_L} \mathcal{H}(\hat{y}_i^L, \mathbf{p}(y|x_i^L)) + \frac{1}{B} \left( \sum_{i=1}^{B_L} \mathcal{H}(\hat{y}_i^L, \mathbf{p}(y|x_i^L)) + \sum_{i=1}^{B_U} \mathcal{H}(\hat{y}_i^U, \mathbf{p}(y|x_i^U)) \right).$$

where $B = B_L + B_U$ and $\alpha_t$ represents the annealing parameter, following the original implementation in Zhu et al. (2024).

We provide comparison of the doubly-robust method with ZeroMatch on Yahoo Answers and Amazon Reviews in Table 19. Our results show that our method outperforms the doubly-robust approach in all settings. We hypothesize that this is due to the asymptotic guarantees of the doubly-robust self-training method failing to hold in the low-label regime.

Table 19: Comaparison with doubly-robust self-training method in Yahoo Answers and Amazon Review with pseudo-labels from FLAN-T5 models. Median and standard deviation of accuracy (%) of 3 different random seeds are reported. Best score among experiments with same pseudo-label set are in bold. 'FM' refers to the foundation model used to generate the pseudo-label set.

| | Dataset | Yahoo Answers | | | Amazon Review | | |
|---|---|---|---|---|---|---|---|
| FM | Label size | 250 | 500 | 2000 | 125 | 250 | 1000 |
| | Doubly-robust | 51.39±12.13 | 44.32±10.91 | 47.9±5.68 | 46.94±0.72 | 47.25±0.63 | 47.76±0.87 |
| A | Zero-shot | 66.63±0.00 | 66.63±0.00 | 66.63±0.00 | 52.37±0.00 | 52.37±0.00 | 52.37±0.00 |
| | ZeroMatch (ours) | **69.78±0.94** | **70.51±0.19** | **71.81±0.07** | **58.83±0.51** | **59.69±0.60** | **60.95±0.13** |
| | Doubly-robust | 24.53±4.84 | 22.52±5.54 | 33.88±2.07 | 35.8±0.13 | 35.9±0.07 | 35.96±0.03 |
| B | Zero-shot | 35.29±0.00 | 35.29±0.00 | 35.29±0.00 | 35.7±0.00 | 35.7±0.00 | 35.7±0.00 |
| | ZeroMatch (ours) | **67.05±0.76** | **67.13±0.73** | **69.61±0.22** | **50.23±1.35** | **53.78±1.74** | **56.12±0.64** |

## C    Additional experiments

### C.1    Strategy to reduce training time overhead

ZeroMatch's KD stage (stage 1) causes around a 50% increase in training time compared with SSL (shown in Table 10). This is due to using the exact same number of training steps from the original SSL algorithm to avoid introducing new hyperparameters in our method. To mitigate this issue, we can use a smaller number of training steps with a larger learning rate to reduce the runtime. In Table 20, we provide experimental results on the Amazon Review dataset with a new setting that reduces the number of KD training steps to 10% and increases the learning rate to 5e-4, and compare with the original ZeroMatch (denoted by KD step ratio 100%) and ZeroMatch without stage 1 (denoted by KD step ratio 0%).

We observe that the new setting (10%) performs close to the original (100%), and still produces a large improvement compared to the case without stage 1 (0%). Note that reducing the KD steps to 10% reduces the overall training runtime overhead of our method from 50% to 5%.

### C.2    Strategy to reduce memory overhead

In AdaMatch, weak augmentations are used only to obtain predictions and do not require storing activations for backpropagation. In contrast, ZeroMatch's auxiliary KD loss in stage 2 involves labeled samples and weakly augmented unlabeled samples, requiring activation storage until backpropagation happens. When

Table 20: Experimental results of reducing KD steps on Amazon Reviews dataset.

| FM | KD step ratio | KD learning rate | Label size | | |
|---|---|---|---|---|---|
| | | | 125 | 250 | 1000 |
| FLAN-T5-XL | 0% | N/A | 55.16±0.30 | 55.75±0.65 | 58.53±0.36 |
| | 10% | 5e-4 | 57.18±0.22 | 59.12±0.52 | 60.31±0.37 |
| | 100% | 1e-4 | **58.83±0.51** | 59.69±0.60 | **60.95±0.13** |
| FLAN-T5-small | 0% | N/A | 45.53±4.91 | 53.19±1.68 | 56.9±0.63 |
| | 10% | 5e-4 | 50.01±2.20 | **54.28±0.84** | **56.49±0.40** |
| | 100% | 1e-4 | **50.23±1.35** | 53.78±1.74 | 56.12±0.64 |

combined with an SSL algorithm, ZeroMatch needs activations for added strongly augmented unlabeled batches for the SSL algorithm, increasing VRAM usage compared to running SSL only (shown in Table 10). To reduce memory usage in ZeroMatch, the auxiliary KD loss in stage 2 can be modified by incorporating strongly augmented samples into the KD loss:

$$\mathcal{L}_{KD_2} = \frac{1}{B}\left(\sum_{i=1}^{B_L}\mathcal{H}(\hat{y}_i^L, \mathbf{q}(y|x_i^L)) + \sum_{i=1}^{B_U}\mathcal{H}(\hat{y}_i^U, \mathbf{q}(y|\mathcal{A}_s(x_i^U)))\right).$$

where $\mathcal{A}_s$ is the same strong augmentations applied to SSL loss discussed in Sec. 3.2. This modification ensures that activations of weakly augmented sample is not stored in our method.

Table 21: Experimental results of ZeroMatch-RM on Amazon Reviews dataset.

| FM | Label size | 125 | 250 | 1000 |
|---|---|---|---|---|
| FLAN-T5-XL | ZeroMatch | 58.83±0.51 | **59.69±0.60** | **60.95±0.13** |
| | ZeroMatch-RM | **58.93±0.66** | 59.50±0.42 | 60.53±0.27 |
| FLAN-T5-small | ZeroMatch | **50.23±1.35** | **53.78±1.74** | 56.12±0.64 |
| | ZeroMatch-RM | 49.23±2.40 | 53.03±0.59 | **56.80±0.86** |

We provide a performance comparison of the new method (denoted by ZeroMatch-RM) on the Amazon Reviews dataset in Table 21. We observe that ZeroMatch-RM performs close to original ZeroMatch while consuming 4.3 GB of memory, which is close to AdaMatch's usage (4.2 GB) in Table 10.

## D    Potential inductive bias mismatch

In our work, we implicitly assumed that pseudo-labels are generated by a reasonably reliable foundation model. These models are typically trained on well-curated datasets to ensure that their outputs follow a meaningful distribution. We assume that pseudo-labels and ground-truth labels have high correlation, and the method may not work well if this is not the case. One possible inductive bias mismatch case is using FM pseudo-labels with added noise (e.g., label flipping with a fixed probability). In such cases, the KD task may shift from learning meaningful representations to memorizing input–noise pairs, which demands significantly more parameter space. This can be an issue in our setup where the backbone model is small, and may lead to significant underperformance compared with SSL.

## E    Limitations and Future Directions

While our work focuses on incorporating pseudo-labels from a single foundation model, future extensions could explore using temperature-controlled logits or using multiple FMs and ensembling their contributions to provide richer supervision.

Because ZeroMatch already includes a knowledge distillation component that mitigates catastrophic forgetting and integrates smoothly with semi-supervised learning, future directions can explore applying our method

to a continual semi-supervised learning setup. For example, when learning a new task, ZeroMatch could combine (1) pseudo-labels generated by the student model trained on previous tasks to reduce forgetting, and (2) new pseudo-labels from foundation models relevant to the new task.

ZeroMatch also naturally extends to semi-supervised domain adaptation settings, since its underlying SSL algorithm (AdaMatch) was designed for domain-shift scenarios. For such settings, it may be beneficial to include mixed batches of both source-domain and target-domain samples during the stage 1 KD phase to help stabilize distillation and improve alignment across domains.

Below are limitations of our work.

- We benchmark our method only on the case where a fixed number of labeled data are sampled per class. This may not demonstrate practical scenarios where labels and classes have imbalanced distributions. Future directions can include benchmarking our method in such settings.

- While we assume that all input data come from a single distribution, practical scenarios may include distribution shifts in the data, especially if collected from sensors on moving edge devices (e.g., cameras in different environments). Future work can explore solutions to adapt to such shifts.

- While most of our experimental results show that our method outperforms the zero-shot method, there is no methodological guarantee that this always holds. Future directions can explore ensuring robust improvement over zero-shot results.

- We also note that our method's performance inherently depends on the quality of the pseudo-labels generated from foundation models. Robustness against labels that are not from foundation models (e.g., weak supervision signals, patterned noise) is not tested and may not work with our method. Future directions can further explore application to different types of label noise.

- We do not include analysis on FM inference API cost. This cost can become a practical issue for large-scale unlabeled data when used with models with high API price or requiring multiple inferences for reasoning. Future directions can explore reducing API cost by using cheaper models for easier samples and requesting high-quality FM predictions only when needed.

