# OpenReview forum: "Enhancing Semi-supervised Learning with Zero-shot Pseudolabels"
_TMLR — Accepted by TMLR_

### Review · Reviewer_vqY2 · 2025-11-20

**Summary Of Contributions:**

ZeroMatch is a two-stage Semi-Supervised Learning (SSL) framework that integrates Knowledge Distillation (KD) with consistency-based SSL. The goal is to train a compact student model effectively on limited labeled data and large amounts of unlabeled data, by leveraging powerful zero-shot pseudo-labels from Foundation Models (FMs). The entire process is designed for low-resource environments, accessing the FM only via a single inference API call.

Pros:
1. The paper is well-formatted and easy to follow.
2. The motivation for using SSL in a low-compute device scenario is interesting.
3. Extensive results show the effectiveness of the proposed method.

Cons:
1. Lack of novelty. The method combines existing common methods (i.e., KD and SSL) and does not propose any novel components or algorithms. It looks more like an engineering solution to me. And it is quite similar to the one[1] mentioned in the related works, but moving to a different setting and making minor modifications.
2. Lack of comparison. There are many SSL methods introduced in the related works but none of them are compared with the proposed method. It casts doubt on the true effectiveness of the proposed method.

[1] Knowledge distillation meets open-set semi-supervised learning, IJCV 2025

Given the above two major concerns, I think the paper does not meet the TMLR standard.

**Audience:**

Yes

**Audience Explanation:**

see above

**Claims And Evidence:**

Yes

**Claims Explanation:**

see above

**Requested Changes:**

see above

---

> ### Author Response · Authors · 2025-12-05
>
> We thank the reviewer for providing helpful feedback. Below are the discussion points to add to weaknesses.
>
> ---
> ## W1. Lack of novelty.
>
> We clarify our contributions in terms of novelty below.
>
> **1. We provide a new motivation to an underexplored problem.**
>
> The problem of using other (teacher) model's pseudo-labels to guide semi-supervised learning was already proposed (see Sec 2.2), but not extensively studied due to the preference for involving teacher model tuning directly into the training process.
> The importance of this problem setting has grown due to recent foundation models having large parameter size, which makes these approaches infeasible when a user does not have access to high-end GPU servers. This new motivation was discussed in Section 5.
>
> **2. We suggest a new approach: combining consistency-based SSL objective with label-based KD.**
>
> While KD and SSL are standard components and there are previous attempts to leverage teacher models and outputs in SSL settings (listed in Sec. 2.2 and 2.3), to the best of our knowledge, no previous attempt was made to combine SSL-specific features (such as strong augmentation, confidence-based sampling) with label-based KD.
>
> ---
> ## W2. ZeroMatch is similar to SRD[1], moving to different settings, with minor modifications.
>
> We clarify key differences between SRD and ZeroMatch (our method) in the table below.
> | Method | SRD | ZeroMatch |
> |--------|-----|-----------|
> | Requires training the teacher model? | Yes | No |
> | Teacher outputs used | Feature representations | Label predictions |
> | Student classifier objective | Supervised learning with only labeled data | Semi-supervised learning with labeled + unlabeled data |
>
> - While we agree that SRD has a similar idea of using auxiliary objectives to match teacher and student output, **SRD's reliance on direct training of the teacher model makes it infeasible for our low-compute setting.**
> - **In low-label settings, ZeroMatch can outperform SRD.**
>     - For SRD, feature representations of the teacher model (pre-trained with labels) can become inaccurate due to the low number of labels, and student model's performance would be upper-bounded by a supervised learning objective (with only labeled data)
>     - ZeroMatch can outperform in this case due to using consistency-based SSL objectives, fully leveraging the unlabeled data.
>
> [1] Knowledge distillation meets open-set semi-supervised learning, IJCV 2025
>
> ---
> ## W3. Lack of comparisons with SSL methods.
>
> We provide a table comparing our method (ZeroMatch) and other SSL methods introduced in the related works section below. The scores for SSL methods are copied from USB[2] which uses the same model and hyperparameter settings as ours.
> **We note that this is not an apples-to-apples comparison.** While SSL methods only use SSL data (labeled and unlabeled), ZeroMatch uses zero-shot pseudo-labels in addition to the SSL data.
>
> Below are results on the CIFAR 100 dataset.
> |FM|SSL method|200|400|
> |--|----------|---|---|
> |None|pi-model|63.76±0.27|73.51±0.64|
> |None|mean teacher|66.84±1.2|74.71±0.67|
> |None|FixMatch|69.55±0.65|80.52±0.93|
> |None|FlexMatch|72.92±0.9|82.33±0.66|
> |None|SimMatch|76.74±1.25|83.18±0.4|
> |CLIP-Base|ZeroMatch|80.87±0.62|82.69±0.23|
> |CLIP-Large|ZeroMatch|85.77±0.05|86.13±0.01|
> |GPT-4.1|ZeroMatch|**87.97±0.20**|**88.12±0.14**|
>
> Below are results on the Amazon reviews dataset.
> |FM|SSL method|250|1000|
> |--|--|--|--|
> |None|pi-model|26.47±6.92|51.73±0.48|
> |None|mean teacher|48.33±0.45|52.49±0.24|
> |None|FixMatch|52.15±1.22|56.27±0.45|
> |None|FlexMatch|54.25±1.21|56.86±0.82|
> |None|SimMatch|52.73±1.73|56.91±0.5|
> |FLAN-T5-small|ZeroMatch|53.78±1.74|56.12±0.64|
> |FLAN-T5-XL|ZeroMatch|59.69±0.6|**60.95±0.13**|
> |GPT-4o|ZeroMatch|**59.82±0.39**|60.19±0.48|
>
> In the results, we observe that our approach combining SSL and FM’s zero-shot pseudo-label significantly outperforms SSL approaches. This highlights the importance of leveraging FM pseudo-labels in an SSL setting when large-scale API inference is available.
>
> We expect ZeroMatch to serve as an add-on for existing SSL approaches to utilize pseudo-labels from foundation models, and discussions on applying ZeroMatch to several SSL methods were provided in Sec 6.5.5.
>
> [2] USB: A Unified Semi-supervised Learning Benchmark for Classification, Neurips 2022

---

### Review · Reviewer_GdrS · 2025-11-25

**Summary Of Contributions:**

This paper introduces ZeroMatch, a framework that combines knowledge distillation from foundation models with semi-supervised learning to simultaneously leverage labeled data, unlabeled data, and zero-shot pseudo-labels. The method trains a compact student model while relying only on inference access to large foundation models, making it suitable for resource-constrained on-device training.

Strengths:
- The paper addresses a timely and interesting problem where users cannot fine-tune large foundation models but can query them for labels.
- The two-stage design is intuitive and experimentally validated across many datasets, with different foundation model types and multiple baselines.
- The method works with closed-source FM APIs, needs minimal VRAM, and reduces data-leakage risks.

Weaknesses:
- While effective, the framework would benefit from a clearer motivation and justification for how the auxiliary head stabilizes training and mitigates forgetting.
- The pseudo-label generation pipeline is detailed in the appendix, but the main text could more clearly explain prompting, calibration, and consistency considerations.
- Stage 1 KD cost is non-trivial: training time increases by ~50%, mitigation strategies could be better discussed.

**Audience:**

Yes

**Audience Explanation:**

Leveraging foundation models in data-scarce settings is a major current topic and the results show clear performance gains without requiring fine tuning, which is attractive.

**Broader Impact Concerns:**

No concern.

**Claims And Evidence:**

Yes

**Claims Explanation:**

The paper provides comprehensive experimental evidence across domains and baselines. Improvements over both semi-supervised learning and foundation models-based zero-shot baselines are consistent and substantial.

**Requested Changes:**

- Expand the explanation of why the auxiliary head mitigates catastrophic forgetting. Provide more analysis of the potential representation drift between the main head and the auxiliary KD head. Clarifying when and how auxiliary supervision helps, or might interfere, would strengthen the paper.
- Discuss computational trade-offs in more detail, including possible strategies to reduce training time or memory overhead.
- Briefly comment on whether ZeroMatch could synergize with related approaches in continual learning or domain adaptation, given its emphasis on preserving knowledge from multiple supervision sources.

---

> ### Author Response · Authors · 2025-12-05
>
> We thank the reviewer for providing insightful comments.
> Below are the discussion points to add to weakness and updates that we have applied to the revision.
>
> ---
> ## W1.1 Clarifying details on auxiliary KD head - why auxiliary head stabilizes training and mitigates catastrophic forgetting.
>
> We added an in-depth explanation of how the auxiliary KD head helps and mitigates catastrophic forgetting in Section 4(in blue text). Below are its contents.
>
> The auxiliary KD head acts as an indirect regularizer that mitigates the catastrophic forgetting issue by providing continuous supervision of the teacher's pseudo-label to the student's model during the SSL training. Because $h_p$ and $h$ share the encoder $g$, the KD loss constrains $g$'s representations to remain aligned with teacher pseudo-labels even as $h$ adapts to the SSL objective. This prevents the encoder from drifting toward erroneous SSL predictions in the early stage of training, which may easily override the teacher's knowledge learned from stage 1.
> $\lambda_p$ and $\alpha_t$ can be set to control the balance between the SSL objective and the KD loss to ensure that the KD loss does not overwhelm the SSL objective but is strong enough to preserve the stage 1 knowledge.
>
> ---
> ## W1.2 Clarifying details on auxiliary KD head - potential representation drift.
>
> We updated our draft including this information in Section 4(in blue text). Below are its contents.
>
> While our method is designed to train an accurate student classifier for a given task, we note that its encoder may have representation drift due to forcing supervision with FM pseudo-labels, which may be different from the student end model’s prediction. For the applications that require embeddings from the encoder, we suggest training a separate student model with an additional distillation stage based on the current student's output.
>
> ---
> ## W1.3 Clarifying details on auxiliary KD head - when and how auxiliary supervision helps, or might interfere.
>
> In section 4, we discussed that the auxiliary KD supervision can be especially helpful in a low-label setting, where an SSL algorithm can develop inaccurate predictions on unlabeled data due to inherently low information in labels, leading to inaccurate predictions that may overwrite the knowledge learned in previous stage.
> Our experimental results show that auxiliary supervision helps improve performance in general (this was discussed in ablation study in Sec 6.5.2)
>
> We added a potential case when auxiliary supervision could interfere in Section 4 (in blue text). Below are its contents:
>
> The auxiliary supervision may interfere with the SSL's objective when $\lambda_p$ is set too large.
> In this case, the KD task may dominate the training objective, reducing the flexibility of the SSL head $h$ to adapt to the downstream task.
>
> ---
> ## W2. Possible strategies to reduce training time.
>
> We updated our paper with a potential solution to increased runtime in Appendix C.1(in blue text). Below are the updated contents.
>
> ZeroMatch's KD stage (stage 1) causes around 50% increase in training time compared with SSL. This is due to using the same number of training steps as the original SSL algorithm, to avoid introducing new hyperparameters for our method.
> To mitigate this issue, **we can use a smaller number of training steps with a larger learning rate**. In the table below, we provide experimental results on Amazon Review dataset with 125, 250, and 1000 labels, with a new setting that reduces the number of KD training steps to 10%, and increases the learning rate. We provide comparison with the original ZeroMatch (denoted by 100% steps ratio) and ZeroMatch without stage 1 (denoted by 0% steps ratio) in Table 8 of paper.
>
> | FM             | KD step ratio | KD lr   | 125            | 250            | 1000           |
> |----------------|----------|---------|----------------|----------------|----------------|
> | **FLAN-T5-XL** | 0%       | N/A     | 55.16±0.30     | 55.75±0.65     | 58.53±0.36     |
> |                | 10%      | 5e-4    | 57.18±0.22     | 59.12±0.52     | 60.31±0.37     |
> |                | 100%     | 1e-4    | **58.83±0.51** | **59.69±0.60**     | **60.95±0.13** |
> | **FLAN-T5-small** | 0%    | N/A     | 45.53±4.91     | 53.19±1.68     | **56.9±0.63**      |
> |                | 10%      | 5e-4    | 50.01±2.20     | **54.28±0.84** | 56.49±0.40 |
> |                | 100%     | 1e-4    | **50.23±1.35**     | 53.78±1.74     | 56.12±0.64     |
>
> We observe that the new setting (10%) can perform close to the original (100%), and still produce a large improvement compared to the case without stage 1 (0%). Note that reducing the KD steps to 10% reduces the overall training runtime overhead of our method from 50% to 5%.

---

> ### Author Response · Authors · 2025-12-05
>
> ## W3. Possible strategies to reduce memory usage
> We added the discussion of a possible solution to reduce memory usage in Appendix C.2 (in blue text). Below are its contents.
>
> The memory overhead of ZeroMatch (compared with SSL) is due to using all the inputs (labeled input, weak-augmented unlabeled input, strong-augmented unlabeled input)  for backpropagation, while SSL does not require backpropagation on the weak-augmented unlabeled input.  A possible solution to remove this overhead is to modify the KD auxiliary loss to match SSL on the inputs used for backpropagation. We denote this method by ZeroMatch-RM and compare with original ZeroMatch in the table below.
>
> |FM|Label size|125|250|1000|
> |--|----------|---|---|----|
> |FLAN-T5-XL|ZeroMatch|58.83±0.51|**59.69±0.60**|**60.95±0.13**|
> ||ZeroMatch-RM|**58.93±0.66**|59.50±0.42|60.53±0.27|
> |FLAN-T5-small|ZeroMatch|**50.23±1.35**|**53.78±1.74**|56.12±0.64|
> ||ZeroMatch-RM|49.23±2.40|53.03±0.59|**56.80±0.86**|
>
> We observe that ZeroMatch-RM performs close to original ZeroMatch while consuming 4.3 GB of memory, which is close to AdaMatch’s usage (4.2 GB).
>
> ---
> ## W4. Can ZeroMatch synergize with related approaches in continual learning or domain adaptation?
>
> We added discussion about how our work could relate to continual learning or domain adaptation as future directions in Appendix E. Below are its contents.
>
> Because ZeroMatch already includes a knowledge-distillation component that mitigates catastrophic forgetting and integrates smoothly with semi-supervised learning, future directions can explore applying our method to a continual semi-supervised learning setup. For example, when learning a new task, ZeroMatch could combine (i) pseudo-labels generated by the student model trained on previous tasks to reduce forgetting, and (ii) new pseudo-labels from foundation models relevant to the new task.
>
> ZeroMatch also naturally extends to semi-supervised domain adaptation settings, since its underlying SSL algorithm (AdaMatch) was designed for domain-shift scenarios. For the domain shift setting, it may be beneficial to include mixed batches of both source-domain and target-domain samples during the stage 1 KD phase, to help stabilize distillation and improve alignment across domains.
>
> ---
> ## W5. Explain pseudo-label generation process in main text.
>
> We updated the paper to include a summary of the generation process in the experiment section(Sec. 6.1, in blue text).
>
> We do not apply any label calibration or filtering in our pseudo-label generation process, as we expect the KD head classifiers can accommodate misaligned label distribution, and manual edits may introduce noise in distilling the teacher's knowledge.

---

### Review · Reviewer_Pb9P · 2025-11-27

**Summary Of Contributions:**

**Summary of the Paper**

This paper introduces ZeroMatch, a semi-supervised learning framework that integrates pseudo-labels from foundation models (FMs) into a two-stage training procedure. First, the student model undergoes knowledge distillation using FM-generated pseudo-labels on both labeled and unlabeled data; second, the model is refined via a standard SSL objective augmented with an auxiliary KD head to mitigate catastrophic forgetting. The method is motivated by a realistic on-device training scenario with strict compute and privacy constraints, where users can access FMs only through inference APIs. Extensive experiments across six datasets (vision and NLP), different label budgets, and multiple pseudo-label sources of varying quality show that ZeroMatch consistently improves over SSL baselines, zero-shot performance, and alternative pseudo-label integration strategies. The paper includes thorough ablations and demonstrates robustness even when pseudo-label quality is low.

**Strengths**
* _Well-motivated problem and clear narrative._ The authors present a compelling, realistic on-device training scenario and articulate why existing approaches (PEFT, prompt engineering, and simple pseudo-supervision) fail under these constraints. The storyline is coherent and easy to follow throughout the paper.
* _Extensive and convincing empirical evaluation._ Experiments span six datasets across NLP and vision, multiple label regimes, and pseudo-label sources ranging from high-quality (GPT-4o/4.1, Llama 3.3-70B) to low-quality (FLAN-T5 B, CLIP-base). The method consistently improves performance, even when the quality of the pseudo-labels is poor. Ablations are comprehensive and include removal of key components, hyperparameter sensitivity, runtime analysis, and compatibility tests with other SSL baselines.
* _Practical, easy-to-integrate method._ The approach only requires adding an auxiliary head and a KD loss; no FM finetuning or gradient access is needed. This makes it practical for settings with inference-only FM access and limited compute.
* _Robustness to pseudo-label noise._ The method shows strong resilience across pseudo-label qualities: even low-quality pseudo-labels rarely degrade performance, and improvements typically grow with the number of labeled samples.

**Weaknesses**
* _Method description lacks important details._ The paper does not clearly specify how pseudo-labels are obtained and mapped into the task label space, especially for vision tasks using GPT-4.1 or CLIP. The mechanics of the auxiliary KD head (e.g., temperature, loss normalization, gradient interactions with the main head) remain vague, limiting reproducibility.
* _Figure 1 is unclear and hard to interpret._ The figure does not visually distinguish teacher and student components, nor does it clearly illustrate the flow of pseudo-labels through the two stages. The role of the auxiliary KD head is particularly hard to infer.
* _Limited discussion of limitations and failure modes._ The method increases training time and requires large-scale API inference over all samples, which may be costly in practice.
* _Presentation issues and minor errors._ The paper contains multiple typos (“a SSL method”, “the both metrics”, “accomodate”, etc.) and occasional grammar issues.

**Audience:**

Yes

**Audience Explanation:**

The paper addresses a timely and practically relevant problem at the intersection of semi-supervised learning and foundation model usage under realistic compute and privacy constraints. Many researchers and practitioners in the TMLR community work with limited labeled data and increasingly rely on large foundation models; understanding how to effectively incorporate FM-generated pseudo-labels into SSL pipelines is of clear interest. The findings are likely to be valuable to a broad subset of the TMLR audience.

**Broader Impact Concerns:**

No major ethical risks are apparent beyond standard concerns with using foundation models. The existing Broader Impact statement is adequate, but it could more explicitly acknowledge that FM-generated pseudo-labels may propagate or amplify model biases into downstream student models. A brief discussion of this risk would strengthen the section.

**Claims And Evidence:**

Yes

**Claims Explanation:**

The paper’s empirical claims are supported by extensive and carefully executed experiments across six datasets, multiple label budgets, and a diverse set of pseudo-label sources. The authors compare against strong SSL baselines, zero-shot performance, and several reasonable pseudo-label integration strategies, and the results consistently support the claimed improvements and robustness. The ablation studies further validate the contribution of each component, and the computational analysis aligns with the stated on-device training motivation. Overall, the evidence provided is thorough, convincing, and consistent with the paper’s claims.

**Requested Changes:**

**Major**
1) Clarify the pseudo-label generation process. Provide a detailed description of how pseudo-labels are produced and mapped to task labels across all modalities (e.g., prompt formats for NLP, label mapping or similarity scoring for vision models like CLIP or GPT-4.1).
2) Improve the explanation of the auxiliary KD head and training dynamics. Elaborate on how the auxiliary KD head interacts with the main classifier (e.g., temperature settings, gradient flow, loss weighting). This part of the method is currently under-specified and needs clearer exposition.
3) Revise Figure 1 for clarity. Clearly distinguish teacher vs. student components, show the flow of pseudo-labels, and better illustrate the two-stage process. The current figure is difficult to interpret.

**Minor**
4) Correct minor typos and improve grammar throughout the text.
5) Discuss limitations and potential failure cases. Briefly describe scenarios where low-quality pseudo-labels might harm performance, and comment on the cost of large one-time FM inference. This will present a more balanced view.
6) Add a short discussion on inductive bias mismatch. Explain how pseudo-labels from large FMs interact with smaller student models and whether label distributions require calibration or filtering.

---

> ### Author Response · Authors · 2025-12-05
>
> We appreciate the reviewer for all the helpful comments. We would like to add to the discussion of weaknesses, and share the updates that we applied to the revision to reflect the reviewer’s comments.
>
> ---
> ## W1. Clarifying details on pseudo-label generation process.
> We updated the paper to include a summary of the generation process in the experiment section(Sec. 6.1, in blue text).
> Note that the details of the pseudo-label generation process were explained in Appendix A.2.
>
> We do not apply any label calibration or filtering, as we expect the KD head classifiers can accommodate misaligned label distribution, and manual edits may introduce noise in distilling the teacher's knowledge.
>
> ---
> ## W2.1. Clarifying details on auxiliary KD head - temperature setting.
> Many KD approaches take temperature settings into account when obtaining knowledge from the teacher model in the form of soft labels(logits). However, our settings assume that teacher FM pseudo-labels are given as one-dimensional classification labels, not softmaxed logits, in order to ensure compatibility with closed-source models where internal logits are not visible to the users.
>
> Involving teacher's logits are definitely an interesting future direction, but we did not include it in the current form to simplify the method and avoid introducing new hyperparameters.
>
> ---
> ## W2.2. Clarifying details on auxiliary KD head - loss weighting
> We note that explanations on the relative weight of the KD loss compared with the SSL objective was already provided in Sec. 4, where we introduce the scaling hyperparameter $\lambda_p$ and annealing parameter $\alpha_t$, and discuss its choice in Sec 6.5.3. Please let us know if you believe further explanations could be useful.
>
> ---
> ## W2.3. Clarifying details on auxiliary KD head - gradient flow
> After the student model learns from the teacher pseudo-labels in stage 1, in stage 2, the student model learns the difference between SSL's ground truth label and the teacher pseudo-label.
> While we agree that clarification based on gradient flow could be helpful, we find it hard to provide a clean explanation for this.
> Instead, **we added an in-depth explanation of how the auxiliary KD head helps training**, in terms of representation learning of the encoder $g$, in Sec. 4 (in blue text).
>
> ---
> ## W3. Issue of increased training time
>
> We updated our paper with a potential solution to increased runtime in Appendix C.1(in blue text). Below are the updated contents.
>
> ZeroMatch's KD stage (stage 1) causes around 50% increase in training time compared with SSL. This is due to using the same number of training steps as the original SSL algorithm, to avoid introducing new hyperparameters for our method.
> To mitigate this issue, **we can use a smaller number of training steps with a larger learning rate**. In the table below, we provide experimental results on Amazon Review dataset with 125, 250, and 1000 labels, with a new setting that reduces the number of KD training steps to 10%, and increases the learning rate. We provide comparison with the original ZeroMatch (denoted by 100% steps ratio) and ZeroMatch without stage 1 (denoted by 0% steps ratio) in Table 8 of paper.
>
> | FM             | KD step ratio | KD lr   | 125            | 250            | 1000           |
> |----------------|----------|---------|----------------|----------------|----------------|
> | **FLAN-T5-XL** | 0%       | N/A     | 55.16±0.30     | 55.75±0.65     | 58.53±0.36     |
> |                | 10%      | 5e-4    | 57.18±0.22     | 59.12±0.52     | 60.31±0.37     |
> |                | 100%     | 1e-4    | **58.83±0.51** | **59.69±0.60**     | **60.95±0.13** |
> | **FLAN-T5-small** | 0%    | N/A     | 45.53±4.91     | 53.19±1.68     | **56.9±0.63**      |
> |                | 10%      | 5e-4    | 50.01±2.20     | **54.28±0.84** | 56.49±0.40 |
> |                | 100%     | 1e-4    | **50.23±1.35**     | 53.78±1.74     | 56.12±0.64     |
>
> We observe that the new setting (10%) can perform close to the original (100%), and still produce a large improvement compared to the case without stage 1 (0%). Note that reducing the KD steps to 10% reduces the overall training runtime overhead of our method from 50% to 5%.

---

> > ### Author Response · Authors · 2025-12-05
> >
> > ## W4. Large-scale API inference over all samples can become costly in practice.
> > We acknowledge that lack of inference cost analysis is a limitation of our work. We added a discussion of this in the limitations section (Appendix E, in blue text)
> > We agree that large-scale API inference cost may become a practical issue, especially for using high-quality models.
> > We did not consider API inference cost in our setting due to following reasons:
> > - FM inference cost decreases every year, thanks to batch optimization techniques and model-architectural innovation.
> > - Users may be in a contract or subscription for using inference services at low cost.
> >
> > Future directions can explore reducing API cost by using cheaper models for simpler samples and requesting high-quality FM predictions on an as-needed basis.
> >
> > ---
> > ## W5. Discuss inductive bias mismatch and potential failure modes.
> >
> > We added a section to discuss potential inductive bias in Appendix D. Below are its contents.
> >
> > In our work, we implicitly assumed that pseudo-labels are generated by a reasonably reliable foundation model. These models are typically trained on well-curated datasets, to ensure that their outputs follow a meaningful distribution. We assume that pseudo-labels and ground truth labels have high correlation, and the method may not work well if this is not the case.
> > One possible such failure mode is using FM pseudo-labels with added noise (e.g. label flipping with fixed probability). In such cases, the KD task may shift from learning meaningful representations to memorizing input-noise pairs, which demands significantly more parameter space. This can be an issue in our setup where the student backbone model is small, and may lead to significant underperformance compared with SSL.
> >
> > ---
> > ## W6. Acknowledge that FM-generated pseudo-labels may propagate model biases into downstream student models.
> >
> > We agree with the reviewer’s comment and added the requested acknowledgement in the broader impact section(in blue text).
> >
> > ---
> > ## W7. Update Figure 1 to clarify teacher / student components and show what the auxiliary KD head is doing.
> >
> > We agree with the reviewer’s suggestion and updated figure 1 in the paper.
> >
> > ---
> > ## W8. Correct Minor typos and improve grammar.
> >
> > We updated the paper to fix the minor typos and grammatical errors.

---

> > > ### Comment · Reviewer_Pb9P · 2025-12-10
> > >
> > > Thank you for the detailed and thoughtful rebuttal, as well as the substantial revisions to the manuscript. All of my earlier questions and concerns have been fully addressed. In particular, the expanded explanation of the pseudo-label generation process, the clearer description of the auxiliary KD head and its role, the added discussion of computational overhead and API-cost limitations, and the new section on potential failure modes significantly improve the clarity and completeness of the work. The updated Figure 1 is also much clearer and effectively illustrates the teacher–student setup and the auxiliary KD head. Overall, these improvements have strengthened the paper and enhanced its readability.

---

> > > > ### Author Response · Authors · 2025-12-10
> > > >
> > > > We thank the reviewer for letting us know the feedback on rebuttal and revision, and again, for all the helpful comments for improving our work.

---

### Decision · Action_Editor_1Dbd · 2025-12-17

**Recommendation:** Accept with minor revision

**Additional Comments:**

For the final version, the authors should ensure that the clarified pseudo-label generation pipeline and auxiliary-head training dynamics are fully integrated into the main text, that computational and API-cost trade-offs are clearly summarized, and that the positioning with respect to closely related KD-SSL work is sharpened to emphasize the distinct setting and contributions. With these refinements, the paper will make a timely contribution of interest to TMLR audience.

**Audience:**

Yes

**Audience Explanation:**

Yes, I think so. The paper focuses on using FoMo's in a semi-supervised setting, comparing it against SSL baselines, all of relevance and interest to the community. All three reviewers are leaning towards the acceptance.

**Claims And Evidence:**

Yes

**Claims Explanation:**

This paper presents a well motivated and interesting framework for integrating foundation model pseudo-labels into semi-supervised learning under realistic on-device / inference-only constraints. The empirical evaluation is extensive, spanning multiple modalities, datasets, label regimes, and pseudo-label qualities, and consistently shows gains over SSL baselines and zero-shot alternatives. Notably, the authors have made a thoughtful effort to address reviewer concerns during revision, significantly improving clarity, reproducability, and balance through expanded method descriptions, new ablations on runtime and memory overhead, more clear articulation of the auxiliary KD head’s role, improved figures, and an explicit discussion of limitations, inductive bias mismatch, and bias propagation risks.